# ADAPTIVE ALGORITHM FOR NON-STATIONARY ONLINE CONVEX-CONCAVE OPTIMIZATION

## ABSTRACT

This paper addresses the problem of Online Convex-Concave Optimization, an extension of Online Convex Optimization to two-player time-varying convex-concave games. Our objective is to minimize the dynamic duality gap (D-DGap), a key performance metric that evaluates the players' strategies against arbitrary comparator sequences. Existing algorithms struggle to achieve optimal performance, particularly in stationary or predictable environments. We propose a novel, modular algorithm comprising three key components: an Adaptive Module that adjusts to varying levels of non-stationarity, a Multi-Predictor Aggregator that selects the optimal predictor from multiple candidates, and an Integration Module that seamlessly combines the strengths of both. Our algorithm guarantees a minimax optimal D-DGap upper bound, up to a logarithmic factor, while also achieving a prediction error-based D-DGap bound. Empirical results further demonstrate the effectiveness and adaptability of the proposed method.

## 1 INTRODUCTION

Online Convex Optimization (OCO, Zinkevich, 2003) is a widely adopted framework for addressing dynamic challenges in various real-world domains, such as online learning (Shalev-Shwartz, 2012), resource allocation (Chen et al., 2017), computational finance (Guo et al., 2021), and online ranking (Chaudhuri & Tewari, 2017). It models repeated interactions between a player and the environment, where at each round $t$, the player selects $x_t$ from a convex set $X$, after which the environment reveals a convex loss function $\ell_t$. The goal is to minimize dynamic regret, defined as the difference between the cumulative loss incurred by the player and that of an arbitrary comparator sequence:

$$\text{D-Reg}_T \coloneqq \sum_{t=1}^{T} \ell_t(x_t) - \sum_{t=1}^{T} \ell_t(u_t), \qquad \forall u_t \in X.$$

According to Zhang et al. (2018), the minimax optimal D-Reg bound is $O(\sqrt{(1+P_T)T})$, where $P_T$ represents the path length of the comparator sequence. Achieving this bound typically relies on the meta-expert framework, which consists of a two-layer structure: the inner layer incorporates multiple experts, each operating a base algorithm with different learning rates, while the outer layer aggregates their advice through a weighted decision-making process. The ADER algorithm, introduced by Zhang et al. (2018), is the first method within this framework to achieve the minimax optimal bound, up to a logarithmic factor. Moreover, certain ADER-like algorithms using implicit updates (Campolongo & Orabona, 2021) or optimistic strategies (Scroccaro et al., 2023) as their base algorithms, can further reduce the D-Reg bound to $O(1)$ in stationary environments or in non-stationary environments with perfect predictability.

Online Convex-Concave Optimization (OCCO) extends the OCO framework by incorporating two interacting players engaged in a sequence of time-varying convex-concave games. This framework is relevant in scenarios such as dynamic pricing (Ferreira et al., 2018) and online advertising auctions (Feng et al., 2023). At round $t$, the two players jointly select a strategy pair $(x_t, y_t)$ from a convex feasible set $X \times Y$, with the $x$-player minimizing and the $y$-player maximizing their respective payoffs, followed by the environment revealing a continuous convex-concave payoff function $f_t$. Both players act without prior knowledge of the current or future payoff functions. Targeting a broad spectrum of non-stationary levels, we introduce the *dynamic duality gap* (D-DGap) as

the performance metric, comparing the players' strategies with an arbitrary comparator sequence in hindsight:

$$\text{D-DGap}_T := \sum_{t=1}^{T} \Big( f_t \left( x_t, v_t \right) - f_t \left( u_t, y_t \right) \Big), \qquad \forall (u_t, v_t) \in X \times Y. \qquad (1)$$

The primary challenge in non-stationary OCCO lies in efficiently adapting to environmental shifts while maintaining a low D-DGap.

The state-of-the-art OCCO algorithm proposed by Zhang et al. (2022b) establishes upper bounds for three metrics: static individual regret (corresponding to the D-DGap with fixed comparators), duality gap (representing the worst-case D-DGap), and dynamic Nash equilibrium regret (addressing intermediate scenarios). However, their algorithm does not guarantee the tightness of these bounds. In fact, the minimax optimal D-DGap upper bound is $O\big(\sqrt{(1 + P_T)T}\big)$, where $P_T = \sum_{t=1}^{T} \left( \|u_t - u_{t-1}\| + \|v_t - v_{t-1}\| \right)$ represents the path length of the comparator sequence. Approximating this minimax optimal bound merely requires each player to independently apply the ADER algorithm. This stems from the fact that OCCO can be interpreted as two interdependent OCO problems, with the D-DGap being equivalent to the sum of two individual D-Regs.

However, applying implicit or optimistic methods to OCCO to further reduce the D-DGap in stationary environments or non-stationary environments with perfect predictions presents challenges. For example, implementing a pair of ADER-like algorithms with optimistic implicit online mirror descent (Scroccaro et al., 2023) as the base algorithms requires the use of predictors $h_t(\,\cdot\,, y_t)$ for the $x$-player and $-h_t(x_t, \,\cdot\,)$ for the $y$-player. This requirement conflicts with the fact that the strategy pair $(x_t, y_t)$ is computed based on the predictor $h_t$, creating a contradiction in the algorithm's structure.

In this paper, we propose a modular algorithm to address these challenges. The algorithm consists three components: the Adaptive Module, the Multi-Predictor Aggregator, and the Integration Module, each serving specific functions:

- Adaptive Module: This module adapts to various levels of non-stationarity, ensuring a minimax optimal D-DGap upper bound of $\widetilde{O}\big(\sqrt{(1 + P_T)T}\big)$. It achieves this by employing a pair of ADER or ADER-like algorithms, which are designed to approximate the minimax optimal D-Reg.
- Multi-Predictor Aggregator: This module enhances decision-making by automatically selecting the most accurate prediction, ensuring a sharp $\widetilde{O}(1)$ D-DGap upper bound in stationary environments or non-stationary environments with perfect predictions. It achieves this by employing the clipped Hedge algorithm.
- Integration Module: This module integrates the capabilities of the Adaptive Module and Multi-Predictor Aggregator, enabling the final strategy to adapt across a wide range of non-stationary levels while effectively tracking the best predictor. It serves as a specialized variant of the meta-expert framework, characterized by the coupling of the meta and expert layers, which necessitates a joint solution.

Our algorithm not only approximates the minimax optimal D-DGap upper bound but also achieves bounds based on prediction error, with any further improvements constrained to at most a logarithmic factor. Specifically, with $d$ available predictors, our algorithm yields

$$\text{D-DGap}_T \leq \widetilde{O} \left( \min \left\{ V_T^1, \, \cdots, \, V_T^d, \, \sqrt{(1 + P_T)T}, \, \sqrt{(1 + C_T)T} \right\} \right),$$

where $V_T^k$ measures the prediction error of the $k$-th predictor, $C_T$ bounds the upper limit of $P_T$.

## 2 RELATED WORK

D-Reg was first introduced by Zinkevich (2003), who demonstrated that greedy projection achieves a D-Reg upper bound of $O\big((1 + P_T)\sqrt{T}\big)$. To approximate the minimax optimal D-Reg of $O\big(\sqrt{(1 + P_T)T}\big)$, Zhang et al. (2018) developed the ADER algorithm, which utilizes the meta-expert framework — a two-layer structure employing multiple learning rates, as illustrated in Meta-Grad (van Erven & Koolen, 2016). Since the introduction of ADER, the meta-expert framework

has effectively addressed various levels of non-stationarity (Lu & Zhang, 2019; Zhao et al., 2020; Zhang, 2020; Zhang et al., 2021; Zhao et al., 2021; Zhang et al., 2022a; Zhao et al., 2022; Lu et al., 2023). To further reduce D-Reg, Campolongo & Orabona (2021) implemented implicit updates, resulting in a D-Reg upper bound driven by the temporal variability of loss functions. Subsequently, Scroccaro et al. (2023) refined this approach by establishing a predictor error-based D-Reg bound using optimistic implicit updates.

OCCO represents a time-varying extension of the minimax problem, which was first introduced by von Neumann (1928). The seminal work of Freund & Schapire (1999) connected the minimax problem to online learning, sparking interest in no-regret algorithms for static environments (Anagnostides et al., 2022; Daskalakis et al., 2015; 2021; Ho-Nguyen & Kılınç-Karzan, 2019; Syrgkanis et al., 2015). Recent research has broadened this focus to time-varying games (Anagnostides et al., 2023; Fiez et al., 2021; Roy et al., 2019), with Cardoso et al. (2018) being the first to explicitly investigate OCCO and introduce the concept of saddle-point regret, later redefined as Nash equilibrium regret (Cardoso et al., 2019). Zhang et al. (2022b) further refined the concept of dynamic Nash equilibrium regret and proposed a parameter-free algorithm that guarantees upper bounds for three metrics: static individual regret, duality gap, and dynamic Nash equilibrium regret. This paper advances Zhang et al. (2022b) by unifying metrics through the D-DGap, ensuring minimax optimality for arbitrary comparators, and further reducing the D-DGap using multiple predictions.

## 3 PRELIMINARIES

Let $\left(\mathscr{X}, \|\cdot\|_{\mathscr{X}}\right)$ and $\left(\mathscr{Y}, \|\cdot\|_{\mathscr{Y}}\right)$ be normed vector spaces. Throughout this paper, we omit norm subscripts if the norm can be easily inferred from the context.

The Fenchel coupling (Mertikopoulos & Sandholm, 2016; Mertikopoulos & Zhou, 2016) induced by a proper function $\varphi$ is defined as $B_\varphi(x, z) \coloneqq \varphi(x) + \varphi^\star(z) - \langle z, x \rangle, \forall (x, z) \in \mathscr{X} \times \mathscr{X}^*$, where $\varphi^\star$ represents the convex conjugate of $\varphi$, given by $\varphi^\star(z) \coloneqq \sup_{x \in \mathscr{X}} \{\langle z, x \rangle - \varphi(x)\}$, and the bilinear map $\langle \cdot, \cdot \rangle \colon \mathscr{X}^* \times \mathscr{X} \to \mathbb{R}$ denotes the canonical dual pairing. Here, $\mathscr{X}^*$ is the dual space of $\mathscr{X}$. Fenchel coupling extends the concept of Bregman divergence to more complex primal-dual settings. According to the Fenchel-Young inequality, we have $B_\varphi(x, z) \geq 0$, with equality holding if and only if $z$ is a subgradient of $\varphi$ at $x$. To simplify notation, we use $x^\varphi$ to denote one such subgradient of $\varphi$ at $x$. By directly applying the definition of Fenchel coupling, we obtain $B_\varphi(x, y^\varphi) + B_\varphi(y, z) - B_\varphi(x, z) = \langle z - y^\varphi, x - y \rangle$. A function $\varphi$ is called $\mu$-strongly convex if $B_\varphi(x, y^\varphi) \geq \frac{\mu}{2} \|x - y\|^2, \forall x, y \in \mathscr{X}$.

The standard simplex refers to the set of all non-negative vectors that sum to 1, defined as $\triangle_d \coloneqq \{\boldsymbol{w} \in \mathbb{R}_+^d \mid \|\boldsymbol{w}\|_1 = 1\}$. The clipped version modifies this by restricting the elements of $\boldsymbol{w}$ to lie within a predefined range, resulting in $\triangle_d^\alpha \coloneqq \{\boldsymbol{w} \in \mathbb{R}_+^d \mid \|\boldsymbol{w}\|_1 = 1, w^i \geq \alpha/d, \forall i = 1, 2, \cdots, d\}$, where $\alpha$ represents the clipping coefficient. The Kullback-Leibler (KL) divergence can be viewed as a specific case of Fenchel coupling, induced by the negative entropy, a 1-strongly convex function. As a result, we have the inequality $\mathrm{KL}(\boldsymbol{w}, \boldsymbol{u}) \geq \frac{1}{2} \|\boldsymbol{w} - \boldsymbol{u}\|_1^2, \forall \boldsymbol{w}, \boldsymbol{u} \in \triangle_d$.

We use big $O$ notation for asymptotic upper bounds and $\widetilde{O}$ to omit polylogarithmic terms.

## 4 MAIN RESULTS

In this section, we first formalize the OCCO framework and outline assumptions. Subsequently, we analyze the Adaptive Module, the Integration Module, and the Multi-Predictor Aggregator in detail. Finally, we elucidate the logical structure of our algorithm and highlight its performance advantages.

### 4.1 PROBLEM FORMALIZATION

OCCO can be formalized as follows: At round $t$,

- $x$-player chooses $x_t \in X$ and $y$-player chooses $y_t \in Y$, where the feasible sets $0 \in X \subset \mathscr{X}$ and $0 \in Y \subset \mathscr{Y}$ are both compact and convex.
- The environment feeds back $f_t \colon X \times Y \to \mathbb{R}$, where $f_t$ is continuous and $f_t(\cdot, y)$ is convex in $X$ for every $y \in Y$ and $f_t(x, \cdot)$ is concave in $Y$ for every $x \in X$.

The goal is to minimize D-DGap. Similar to previous studies in online learning, we introduce the following standard assumptions.

**Assumption 1.** *The diameter of $X$ is denoted as $D_X$, and the diameter of $Y$ is denoted as $D_Y$.*

**Assumption 2.** *All payoff functions are bounded, and their subgradients are also bounded. Specifically, $\exists M$, $G_X$ and $G_Y$, such that $\forall x \in X$, $\forall y \in Y$ and $\forall t$, the following inequalities hold:*

$$|f_t(x, y)| \leq M, \qquad \|\nabla_x f_t(x, y)\| \leq G_X, \qquad \|\nabla_y(-f_t)(x, y)\| \leq G_Y.$$

### 4.2 Adaptive Module

In algorithms that adapt automatically to varying levels of non-stationarity, existing methods generally rely on the meta-expert framework to approximate the minimax optimal D-Reg. The ADER algorithm serves as an example of this framework, with variants that include replacing the base algorithm with implicit updates (Campolongo & Orabona, 2021) or optimistic strategies (Scroccaro et al., 2023). The following proposition illustrates the results achieved by decomposing the OCCO problem into two OCO problems and independently executing two ADER or ADER-like algorithms.

**Proposition 1.** *Consider running two ADER or ADER-like algorithms independently, both designed to approximate the minimax optimal D-Reg. In round $t$, one algorithm outputs $\overline{x}_t$, receiving the convex loss function $f_t(\cdot, y_t)$, while the other generates $\overline{y}_t$, receiving $-f_t(x_t, \cdot)$. Under Assumptions 1 and 2, this setup yields the following inequality for all $(u_t, v_t) \in X \times Y$:*

$$\sum_{t=1}^{T} \Big( f_t(x_t, v_t) - f_t(x_t, \overline{y}_t) \Big) + \sum_{t=1}^{T} \Big( f_t(\overline{x}_t, y_t) - f_t(u_t, y_t) \Big) \leq \widetilde{O}\left( \sqrt{(1 + \min\{P_T, C_T\}) T} \right),$$

*where $P_T = \sum_{t=1}^{T} \left( \|u_t - u_{t-1}\| + \|v_t - v_{t-1}\| \right)$, $C_T = \sum_{t=1}^{T} \left( \|x'_t - x'_{t-1}\| + \|y'_t - y'_{t-1}\| \right)$ bounds the upper limit of $P_T$, $x'_t = \arg\min_{x \in X} f_t(x, y_t)$ and $y'_t = \arg\max_{y \in Y} f_t(x_t, y)$.*

The two ADER or ADER-like algorithms outlined in Proposition 1 operate independently, with each algorithm's output influencing the other's loss function. This mutual dependence introduces challenges in further tightening the upper bound of the D-DGap, particularly in favorable environments such as stationary or predictable scenarios. In the OCCO setting, the two players can jointly formulate strategies to more effectively respond to environmental changes. As a result, we do not use the output $(\overline{x}_t, \overline{y}_t)$ from Proposition 1 as the final strategy. Instead, we treat the method in Proposition 1 as an *Adaptive Module*, designed to handle various levels of non-stationarity. Its outputs provide recommendations that adapt to uncertainties in the path length of the comparator sequence. In the following section, we explore how the two players can collaboratively update their strategies.

Before concluding this section, we state that the D-DGap upper bound, as presented in Proposition 1, is minimax optimal up to a logarithmic factor. The following proposition guarantees this conclusion.

**Proposition 2** (D-DGap Lower Bound)**.** *Regardless of the strategies adopted by the players, there always exists a sequence of convex-concave payoff functions satisfying Assumption 2, and a comparator sequence satisfying $P_T \leq P$, ensuring a D-DGap of at least $\Omega\left( \sqrt{(1 + \min\{P, C_T\})T} \right)$.*

Proposition 2 can be intuitively justified. Specifically, the D-DGap is essentially the sum of two individual D-Regs. According to Theorem 2 of Zhang et al. (2018), in adversarial environments, no online algorithm can bound the individual D-Regs below $\Omega\left( \sqrt{(1 + P^u)T} \right)$ and $\Omega\left( \sqrt{(1 + P^v)T} \right)$, respectively, where $P^u \geq \sum_{t=1}^{T} \|u_t - u_{t-1}\|$, and $P^v \geq \sum_{t=1}^{T} \|v_t - v_{t-1}\|$. Consequently, the D-DGap lower bound cannot be less than the sum of these regret lower bounds. Since the lower bound depends on $P$, a preset upper limit for $P_T$, setting $P$ above $C_T$ would render the bound overly loose, making $C_T$ the effective threshold. This reasoning validates Proposition 2.

### 4.3 Integration Module

Our goal is to design an algorithm that not only adapts automatically to arbitrary comparator sequences but also achieves a prediction error-based D-DGap upper bound. To this end, we propose a specialized variant of the meta-expert framework, where the expert layer consists of two key experts:

- The first expert is tailored to ensure the algorithm achieves a D-DGap upper bound driven by prediction errors, playing a crucial role in tightening the D-DGap in predictable environments.

- The second expert is an Adaptive Module that adjusts dynamically to any sequence of comparators, guaranteeing minimax optimality across a broad spectrum of non-stationarity levels.

The core innovation of this framework lies in its joint update mechanism, where the first expert and the meta layer are updated in coordination. This design allows the framework to seamlessly incorporate the first expert's insights into the final strategy, ensuring both adaptability and precise control over the D-DGap upper bound.

We refer to this framework as the *Integration Module*. Below, we provide a comprehensive analysis of its implementation.

Let $(\widehat{x}_t, \widehat{y}_t)$ represent the advice from the first expert, and $(\overline{x}_t, \overline{y}_t)$ denote the strategy pair generated by the second expert. The final strategies of the Integration Module are defined as $x_t = [\widehat{x}_t, \overline{x}_t] \boldsymbol{w}_t$ and $y_t = [\widehat{y}_t, \overline{y}_t] \boldsymbol{\omega}_t$, where $\boldsymbol{w}_t$ and $\boldsymbol{\omega}_t$ are the weights provided by the meta layer. Since the second expert's advice $(\overline{x}_t, \overline{y}_t)$ is determined by the Adaptive Module, the Integration Module must internally generate both the first expert's advice $(\widehat{x}_t, \widehat{y}_t)$ and the weight parameters $\boldsymbol{w}_t$ and $\boldsymbol{\omega}_t$.

Let the arbitrary convex-concave predictor $h_t$ serve as a hint for the two players. Define

$$\boldsymbol{A}_t = \begin{bmatrix} f_t(\widehat{x}_t, \widehat{y}_t), & f_t(\widehat{x}_t, \overline{y}_t) \\ f_t(\overline{x}_t, \widehat{y}_t), & f_t(\overline{x}_t, \overline{y}_t) \end{bmatrix}, \qquad \boldsymbol{\Lambda}_t = \begin{bmatrix} h_t(\widehat{x}_t, \widehat{y}_t), & h_t(\widehat{x}_t, \overline{y}_t) \\ h_t(\overline{x}_t, \widehat{y}_t), & h_t(\overline{x}_t, \overline{y}_t) \end{bmatrix},$$

and let $\boldsymbol{w} = [w, 1-w]^{\mathrm{T}}, \boldsymbol{\omega} = [\omega, 1-\omega]^{\mathrm{T}}$,

$$H_t(x, y; \boldsymbol{w}, \boldsymbol{\omega}) = \boldsymbol{w}^{\mathrm{T}} \begin{bmatrix} h_t(x, y), & h_t(x, \overline{y}_t) \\ h_t(\overline{x}_t, y), & h_t(\overline{x}_t, \overline{y}_t) \end{bmatrix} \boldsymbol{\omega} + \frac{w}{\eta_t} B_\phi(x, \widetilde{x}_t^\phi) - \frac{\omega}{\gamma_t} B_\psi(y, \widetilde{y}_t^\psi),$$

$$W_t(\boldsymbol{w}, \boldsymbol{\omega}; x, y) = \boldsymbol{w}^{\mathrm{T}} \begin{bmatrix} h_t(x, y), & h_t(x, \overline{y}_t) \\ h_t(\overline{x}_t, y), & h_t(\overline{x}_t, \overline{y}_t) \end{bmatrix} \boldsymbol{\omega} + \frac{1}{\theta_t} \mathrm{KL}(\boldsymbol{w}, \widetilde{\boldsymbol{w}}_t) - \frac{1}{\vartheta_t} \mathrm{KL}(\boldsymbol{\omega}, \widetilde{\boldsymbol{\omega}}_t).$$

(2)

The Integration Module can then be represented by the following updates:

$$\text{First Expert:} \begin{cases} (\widehat{x}_t, \widehat{y}_t) = \arg\min_{x \in X} \max_{y \in Y} H_t(x, y; \boldsymbol{w}_t, \boldsymbol{\omega}_t), & \text{(3a)} \\ \widetilde{x}_{t+1} = \arg\min_{x \in X} \eta_t [f_t(x, \widehat{y}_t), f_t(x, \overline{y}_t)] \boldsymbol{\omega}_t + B_\phi(x, \widetilde{x}_t^\phi), & \text{(3b)} \\ \widetilde{y}_{t+1} = \arg\max_{y \in Y} \gamma_t [f_t(\widehat{x}_t, y), f_t(\overline{x}_t, y)] \boldsymbol{w}_t - B_\psi(y, \widetilde{y}_t^\psi), & \text{(3c)} \end{cases}$$

$$\text{Meta Layer:} \begin{cases} (\boldsymbol{w}_t, \boldsymbol{\omega}_t) = \arg\min_{\boldsymbol{w} \in \triangle_2^\alpha} \max_{\boldsymbol{\omega} \in \triangle_2^\alpha} W_t(\boldsymbol{w}, \boldsymbol{\omega}; \widehat{x}_t, \widehat{y}_t), & \text{(3d)} \\ \widetilde{\boldsymbol{w}}_{t+1} = \arg\min_{\boldsymbol{w} \in \triangle_2^\alpha} \theta_t \boldsymbol{w}^{\mathrm{T}} \boldsymbol{A}_t \boldsymbol{\omega}_t + \mathrm{KL}(\boldsymbol{w}, \widetilde{\boldsymbol{w}}_t), & \text{(3e)} \\ \widetilde{\boldsymbol{\omega}}_{t+1} = \arg\max_{\boldsymbol{\omega} \in \triangle_2^\alpha} \vartheta_t \boldsymbol{w}_t^{\mathrm{T}} \boldsymbol{A}_t \boldsymbol{\omega} - \mathrm{KL}(\boldsymbol{\omega}, \widetilde{\boldsymbol{\omega}}_t), & \text{(3f)} \end{cases}$$

where $\eta_t, \gamma_t, \theta_t$ and $\vartheta_t$ are learning rates, and $\alpha = 2/T$. To facilitate our analysis, we assume that the regularizers $\phi$ and $\psi$ are both 1-strongly convex, and their Fenchel couplings satisfy Lipschitz continuity with respect to the first variable, i.e., $\exists L_\phi, L_\psi < +\infty, \forall \alpha, x, x' \in X, \forall \beta, y, y' \in Y$:

$$\left| B_\phi(x, \alpha^\phi) - B_\phi(x', \alpha^\phi) \right| \leq L_\phi \|x - x'\|, \qquad \left| B_\psi(y, \beta^\psi) - B_\psi(y', \beta^\psi) \right| \leq L_\psi \|y - y'\|.$$

These assumptions are consistent with previous literature (Campolongo & Orabona, 2021; Zhang et al., 2022b).

As Equations (3a) and (3d) necessitate a joint solution, we first postpone the discussion of the solution methodology and focus on how the above updates implement the functionality of the Integration Module.

The following two theorems describe the performance of the first expert and the meta layer in the Integration Module. The specifications for the adaptive learning rates align with the methodology outlined by Campolongo & Orabona (2021). Refer to the full versions of the theorems in the appendix for details.

**Theorem 3** (Performance Guarantee for the First Expert). *Under Assumptions 1 and 2. If $\eta_t$ and $\gamma_t$ follow adaptive learning rates, then the following inequality holds:*

$$\sum_{t=1}^{T} \left( f_t(x_t, v_t) - \boldsymbol{w}_t^{\mathrm{T}} \boldsymbol{A}_t^{:,1} \right) + \sum_{t=1}^{T} \left( \boldsymbol{A}_t^{1,:} \boldsymbol{\omega}_t - f_t(u_t, y_t) \right) \leq O \left( \sum_{t=1}^{T} \rho(f_t, h_t) \right),$$

*where $\rho(f_t, h_t) = \max_{x \in X, y \in Y} |f_t(x, y) - h_t(x, y)|$ measures the distance between $f_t$ and $h_t$.*

**Theorem 4** (Performance Guarantee for the Meta Layer). *Under Assumption 2, and assume that $T \geq 2$. If $\theta_t$ and $\vartheta_t$ follow adaptive learning rates, then the meta layer of the Integration Module enjoys the following inequality:*

$$\sum_{t=1}^{T} \left( \boldsymbol{w}_t^{\mathrm{T}} \boldsymbol{A}_t \boldsymbol{v} - \boldsymbol{u}^{\mathrm{T}} \boldsymbol{A}_t \boldsymbol{\omega}_t \right) \leq O \left( \min \left\{ \sum_{t=1}^{T} \rho(f_t, h_t), \ \sqrt{(1 + \ln T)T} \right\} \right), \qquad \forall \boldsymbol{u}, \boldsymbol{v} \in \triangle_2.$$

The following theorem provides performance guarantee for the Integration Module.

**Theorem 5** (D-DGap for the Integration Module). *Under the settings of Proposition 1 and Theorems 3 and 4, the Integration Module achieves the following D-DGap bound:*

$$\text{D-DGap}_T \leq \widetilde{O} \left( \min \left\{ \sum_{t=1}^{T} \rho(f_t, h_t), \ \sqrt{\left( 1 + \min\{P_T, C_T\} \right) T} \right\} \right).$$

*Proof of Theorem 5.* Following the first expert, the D-DGap can be equivalently transformed into

$$\sum_{t=1}^{T} \left( \left( f_t(x_t, v_t) - \boldsymbol{w}_t^{\mathrm{T}} \boldsymbol{A}_t^{:,1} \right) + \left( \boldsymbol{w}_t^{\mathrm{T}} \boldsymbol{A}_t \begin{bmatrix} 1 \\ 0 \end{bmatrix} - [1, 0] \boldsymbol{A}_t \boldsymbol{\omega}_t \right) + \left( \boldsymbol{A}_t^{1,:} \boldsymbol{\omega}_t - f_t(u_t, y_t) \right) \right).$$

By applying Theorems 3 and 4, the resulting bound is $\text{D-DGap}_T \leq O\left( \sum_{t=1}^{T} \rho(f_t, h_t) \right)$. On the other hand, following the second expert, the upper bound for the D-DGap is

$$\sum_{t=1}^{T} \left( \left( f_t(x_t, v_t) - f_t(x_t, \overline{y}_t) \right) + \left( \boldsymbol{w}_t^{\mathrm{T}} \boldsymbol{A}_t \begin{bmatrix} 0 \\ 1 \end{bmatrix} - [0, 1] \boldsymbol{A}_t \boldsymbol{\omega}_t \right) + \left( f_t(\overline{x}_t, y_t) - f_t(u_t, y_t) \right) \right).$$

Applying Proposition 1 and Theorem 4 yields $\text{D-DGap}_T \leq \widetilde{O}\left( \sqrt{(1 + \min\{P_T, C_T\})T} \right)$. Combining the two results above yields the desired conclusion. $\qquad\square$

The rationale for designing Equations (3a) and (3d) as a joint update strategy is as follows: To guarantee that the final D-DGap attains a prediction error-based bound, both the first expert and the meta layer must achieve such bounds independently. This objective requires mutual dependence: the first expert needs to be aware of the meta-layer's weights to update its advice, while the meta layer relies on advice from both experts to adjust its own weights. This inherent interdependence necessitates a joint solution.

After analyzing the functionality of the Integration Module, we now turn to the methodology for jointly solving Equations (3a) and (3d). The following theorem not only asserts the existence of solutions to this problem but also suggests methods for solving it.

**Theorem 6.** *Let $\mathscr{H}_t \colon (\boldsymbol{w}, \boldsymbol{\omega}) \mapsto (\widehat{x}, \widehat{y})$, where $(\widehat{x}, \widehat{y})$ is the saddle point of $H_t(\cdot, \cdot; \boldsymbol{w}, \boldsymbol{\omega})$, and let $\mathscr{W}_t \colon (\widehat{x}, \widehat{y}) \mapsto (\boldsymbol{w}, \boldsymbol{\omega})$, where $(\boldsymbol{w}, \boldsymbol{\omega})$ is the saddle point of $W_t(\cdot, \cdot; \widehat{x}, \widehat{y})$. Then, the composition map $\mathscr{W}_t \circ \mathscr{H}_t$ has a fixed point, which corresponds to the solution of Equations (3a) and (3d). Furthermore, let $\sigma = \max_{\boldsymbol{v} \in \triangle_2^{\alpha}} W_t(\boldsymbol{w}, \boldsymbol{v}; \widehat{x}, \widehat{y}) - \min_{\boldsymbol{u} \in \triangle_2^{\alpha}} W_t(\boldsymbol{u}, \boldsymbol{\omega}; \widehat{x}, \widehat{y})$, and denote $\boldsymbol{w} = [w, 1 - w]^{\mathrm{T}}$, $\boldsymbol{\omega} = [\omega, 1 - \omega]^{\mathrm{T}}$. Then the map $\mathscr{F}_t \colon (w, \omega) \mapsto \sigma$ is continuous from the closed rectangular region $[1/T, 1 - 1/T]^2$ to $\mathbb{R}_{\geq 0}$, and solving the fixed point of $\mathscr{W}_t \circ \mathscr{H}_t$ is equivalent to minimizing $\mathscr{F}_t$ to zero.*

Now solving Equations (3a) and (3d) is equivalent to locating a point within the closed rectangular region $[1/T, 1 - 1/T]^2$ that reduces the continuous map $\mathscr{F}_t$ to zero. Various methods can address this minimization problem, including Particle Swarm Optimization (PSO, Kennedy & Eberhart, 1995) or Successive Reduction of Search Space within feasible computational limits. As an illustration, we provide a PSO-based approach (see Algorithm 1), where the PSO iteration can be guided by Shi & Eberhart (1998) or Sun et al. (2004).

The efficiency of the computation is largely dependent on the saddle-point solver, particularly at line 6 of Algorithm 1. Due to the specific structure of the payoff functions, a universal solution is often unattainable. In some cases, closed-form solutions are feasible, especially when the regularizers $\phi$ and $\psi$ are quadratic (e.g., squared Euclidean norms) and the payoff function $f_t$ is bilinear or quadratic, which can significantly reduce computational costs. When closed-form solutions are not available, numerical methods such as those demonstrated by Abernethy et al. (2018); Carmon et al. (2020); Jin et al. (2022) have proven effective, offering both fast and reliable convergence.

---

**Algorithm 1** Particle Swarm Optimization for Solving Equations (3a) and (3d)

---

**Input:** $H_t$ and $W_t$ according to Equation (2)
1: Initialize $N$ particles, each with a random position $(w^i, \omega^i)$ in $[1/T, 1 - 1/T]^2$, $i = 1, \cdots, N$
2: **repeat**
3:     **for** each particle $i = 1, 2, \cdots, N$ **do**
4:         Update position $(w^i, \omega^i)$ using Particle Swarm Optimization iteration
5:         $\boldsymbol{w}^i \leftarrow [w^i, 1 - w^i]^{\mathrm{T}}$ and $\boldsymbol{\omega}^i \leftarrow [\omega^i, 1 - \omega^i]^{\mathrm{T}}$
6:         Calculate the saddle point $(\widehat{x}^i, \widehat{y}^i) = \arg\min_{x \in X} \max_{y \in Y} H_t\left(x, y; \boldsymbol{w}^i, \boldsymbol{\omega}^i\right)$
7:         Get the duality gap $\sigma^i = \max_{\boldsymbol{v} \in \triangle_2^\alpha} W_t\left(\boldsymbol{w}^i, \boldsymbol{v}; \widehat{x}^i, \widehat{y}^i\right) - \min_{\boldsymbol{u} \in \triangle_2^\alpha} W_t\left(\boldsymbol{u}, \boldsymbol{\omega}^i; \widehat{x}^i, \widehat{y}^i\right)$
8:         Update $\sigma_{\text{best}}^i$, the best known duality gap of particle $i$
9:     **end for**
10:    Update global best known duality gap $\sigma_{\text{gbest}}$, and record the corresponding particle index $j$
11: **until** $\sigma_{\text{gbest}} \downarrow 0$
**Output:** $\widehat{x}_t \leftarrow \widehat{x}^j, \widehat{y}_t \leftarrow \widehat{y}^j, \boldsymbol{w}_t \leftarrow \boldsymbol{w}^j, \boldsymbol{\omega}_t \leftarrow \boldsymbol{\omega}^j$

---

### 4.4 Multi-Predictor Aggregator

The output of the integrated module achieves minimax optimality and effectively reduces the D-DGap when using an accurate predictor sequence. However, relying on a single predictor sequence limits the algorithm's adaptability to different environments. To address this, we consider having $d$ available predictor sequences, each potentially derived from distinct models of the underlying environment. Our goal is to enhance the Integration Module by supporting multiple predictors, enabling it to retain minimax optimality while dynamically adapting to the most effective predictor sequence across these models.

We propose the following *Multi-Predictor Aggregator*: At each round $t$, there are $d$ available predictors, denoted as $\{h_t^1, h_t^2, \cdots, h_t^d\}$. The Multi-Predictor Aggregator outputs an aggregated predictor $h_t = \sum_{k=1}^d \xi_t^k h_t^k$ to the Integration Module. The weight vector $\boldsymbol{\xi}_t = [\xi_t^1, \xi_t^2, \cdots, \xi_t^d]^{\mathrm{T}}$ is derived using the clipped Hedge algorithm, which solves the following optimization problem:

$$\boldsymbol{\xi}_{t+1} = \arg\min_{\boldsymbol{\xi} \in \triangle_d^a} \zeta_t \langle \boldsymbol{L}_t, \boldsymbol{\xi} \rangle + \mathrm{KL}(\boldsymbol{\xi}, \boldsymbol{\xi}_t), \tag{4}$$

where $a = d/T$, $\zeta_t$ is the learning rate, and $\boldsymbol{L}_t$ denotes the loss vector:

$$\boldsymbol{L}_t = [L_t^1, L_t^2, \cdots, L_t^d]^{\mathrm{T}}, \qquad L_t^k = \max_{x \in \{\widehat{x}_t, \overline{x}_t, \widetilde{x}_{t+1}\}, y \in \{\widehat{y}_t, \overline{y}_t, \widetilde{y}_{t+1}\}} \left| f_t(x, y) - h_t^k(x, y) \right|.$$

The following theorem states that the Multi-Predictor Aggregator effectively provides multiple predictor support for the Integration Module. The specification for the adaptive learning rate follows the methodology described by Campolongo & Orabona (2021), as detailed in Theorem 10 in the appendix.

**Theorem 7** (D-DGap for the Integration Module Using a Multi-Predictor Aggregator). *Assume the payoff function $f_t$ and all predictors $\{h_t^1, h_t^2, \cdots, h_t^d\}$ satisfy Assumption 2. Let $T \geq d$. If the learning rate $\zeta_t$ of Multi-Predictor Aggregator follows the adaptive rule, then the D-DGap upper bound for the Integration Module can be enhanced as follows:*

$$\text{D-DGap}_T \leq \widetilde{O}\left(\min\left\{\min_{k \in \{1, 2, \cdots, d\}} \sum_{t=1}^T \rho(f_t, h_t^k), \sqrt{(1 + \min\{P_T, C_T\})T}\right\}\right).$$

The rationale for using the Hedge algorithm is its ability to perform consistently close to the best expert's strategy over time, making it an effective choice when multiple predictors are involved. An efficient solution for the clipped Hedge can be found in Figure 3 of Herbster & Warmuth (2001).

### 4.5 Structure and Advantages

In the previous subsections, we analyzed the Adaptive Module, Integration Module, and Multi-Predictor Aggregator individually. To clarify how these modules work together to form the overall

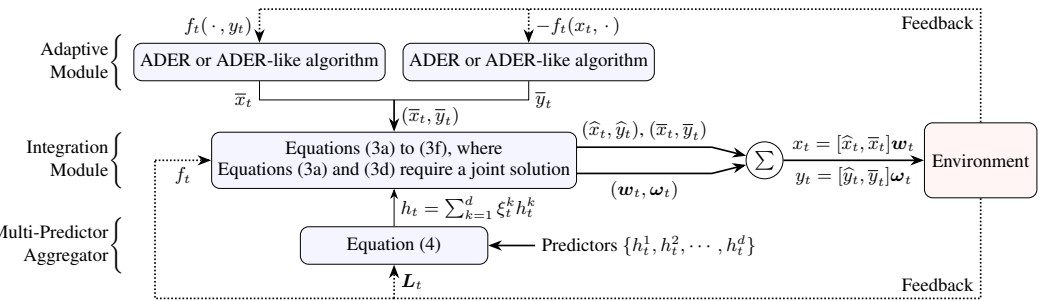

Figure 1: Structural Diagram of Our Algorithm.

---

**Algorithm 2** Pseudocode for Our Algorithm

---

**Require:** $X$ and $Y$ satisfy Assumption 1. All payoff functions and predictors satisfy Assumption 2
**Initialize:** $\widetilde{x}_1, \widetilde{y}_1, \widetilde{\boldsymbol{w}}_1, \widetilde{\boldsymbol{\omega}}_1, \boldsymbol{\xi}_1$ and $(\overline{x}_1, \overline{y}_1)$

1: **for** $t = 1, 2, \cdots, T$ **do**
2:     Receive $d$ predictors $h_t^1, h_t^2, \cdots, h_t^d$ and compute $h_t = \sum_{k=1}^d \xi_t^k h_t^k$
3:     Obtain $(\widehat{x}_t, \widehat{y}_t)$ and $(\boldsymbol{w}_t, \boldsymbol{\omega}_t)$ by jointly updating Equations (3a) and (3d)
4:     Output $x_t = [\widehat{x}_t, \overline{x}_t]\boldsymbol{w}_t$, $y_t = [\widehat{y}_t, \overline{y}_t]\boldsymbol{\omega}_t$, and then observe $f_t$
5:     Update $\widetilde{x}_{t+1}, \widetilde{y}_{t+1}, \widetilde{\boldsymbol{w}}_{t+1}$ and $\widetilde{\boldsymbol{\omega}}_{t+1}$ according to Equations (3b), (3c), (3e) and (3f)
6:     Update $\boldsymbol{\xi}_{t+1}$ according to Equation (4)
7:     Update $(\overline{x}_{t+1}, \overline{y}_{t+1})$ by running two ADER or ADER-like algorithms
8: **end for**

---

algorithm, we present a structural diagram (see Figure 1) and accompanying pseudocode (see Algorithm 2).

Theorem 7 provides the D-DGap upper bound guarantee for the entire algorithm, which can be rearranged as follows:

$$\text{D-DGap}_T \leq \widetilde{O}\Big(\min\Big\{\underbrace{\min\{V_T^1, \cdots, V_T^d\}}_{(5a)}, \underbrace{\sqrt{(1 + \min\{P_T, C_T\})T}}_{(5b)}\Big\}\Big), \tag{5}$$

where $V_T^k = \sum_{t=1}^T \rho(f_t, h_t^k)$ represents the prediction error of the $k$-th predictor.

The Adaptive Module establishes a minimax optimal bound as indicated in Equation (5b), enabling the algorithm to effectively adapt to varying levels of non-stationarity. Concurrently, the Multi-Predictor Aggregator contributes the bound described in Equation (5a). When any single predictor accurately models the environment, the algorithm attains a sharp $\widetilde{O}(1)$ D-DGap, effectively functioning as an automatic selection mechanism for the best predictor among the available options.

The Integration Module combines the strengths of the Adaptive Module and the Multi-Predictor Aggregator, ensuring that the final strategy is both highly adaptive to non-stationary settings and capable of efficiently tracking the most accurate predictor. This integration guarantees near-optimal performance in diverse scenarios, with any potential improvement limited to at most a logarithmic factor.

## 5 EXPERIMENTS

This section validates the effectiveness of our algorithm through experimental evaluation. We choose the algorithm proposed by Zhang et al. (2022b) as the benchmark for comparison.

We consider a specific instance of the OCCO problem, where the feasible domain is defined as $X \times Y = [-1, 1]^2$, and the environment provides the following convex-concave payoff function at round $t$:

$$f_t(x, y) = \frac{1}{2}(x - x_t^*)^2 - \frac{1}{2}(y - y_t^*)^2 + (x - x_t^*)(y - y_t^*),$$

Table 1: Four Environment Settings. In this table, the saddle point $(x_t^*, y_t^*)$ is expressed in the complex form $p_t^* = x_t^* + iy_t^*$, where $i$ is the imaginary unit, satisfying $i^2 = -1$. $z_1(t) = \ln(1+t)$, $z_2(t) = \ln\ln(e+t)$. As $t$ increases, the growth rates of both $z_1$ and $z_2$ gradually decelerate. $\varepsilon \sim U(0,1)$ is random variable that follows a uniform distribution on the interval $[0,1]$, and $\varphi \sim N(\pi, 1)$ is a random angle that follows a Gaussian distribution with mean $\pi$.

| Case | I | II | III | IV |
|---|---|---|---|---|
| $x_t^* + iy_t^*$ | $\frac{1}{3}z_2(t)e^{iz_1(t)}$ | $\frac{1}{3}z_2(t)e^{i\frac{2\pi}{3}t+iz_2(t)}$ | $\frac{1}{2}e^{\frac{1}{7}(\varepsilon+i2\pi t)}$ | $\frac{1}{2}e^{i(\varphi+\arg(x_t+iy_t))}$ |
| Trajectories |  |  |  |  |
| Property | $\rho(f_t, f_{t-1}) \to 0$ | $\rho(f_t, f_{t-3}) \to 0$ | $\left\|p_t^* - p_{t-7}^*\right\| \le \frac{1-e^{1/7}}{2}$ | Adversarial |

Table 2: Three Levels on Comparator Sequence Non-Stationarity.

| Level | i | ii | iii |
|---|---|---|---|
| Comparator | $(u_t, v_t) \equiv (0,0)$ | $(u_t, v_t) = (x_t^*, y_t^*)/\ln(1+t)$ | $(u_t, v_t) = (x_t', y_t')$ |

where $(x_t^*, y_t^*) \in X \times Y$ denotes the saddle point of $f_t$. This setup satisfies Assumptions 1 and 2. The evolution of the saddle point $(x_t^*, y_t^*)$ reflects specific environmental characteristics. We identify four distinct cases, as outlined in Table 1:

- Case I indicates a gradually stationary environment, with the movement of the saddle point diminishing over time.
- Case II and III represent approximate periodic environments. In Case II, the saddle point cycles among three branches, while in Case III, it cycles among seven, with its position in each branch chosen randomly.
- Case IV depicts an adversarial environment where the saddle point cannot be effectively approximated. In this case, upon selecting a strategy pair $(x_t, y_t)$, the environment generates the saddle point $(x_t^*, y_t^*)$ by rotating the strategy pair by a random angle $\varphi \sim N(\pi, 1)$ and then projecting it onto the circle of radius $1/2$.

To capture a range of non-stationarity levels, we select three comparator sequences representing different dynamics, from fully stationary to highly non-stationary settings, as detailed in Table 2.

We instantiate our algorithm as follows: Let $\phi(x) = x^2/2$ and $\psi(y) = y^2/2$. Both $B_\phi$ and $B_\psi$ are bounded, 1-strongly convex, and exhibit Lipschitz continuity with respect to their first variables. For the Multi-Predictor Aggregator, we configure four predictors: $h_t^1 = f_{t-5}$, $h_t^2 = f_{t-6}$, $h_t^3 = f_{t-7}$, and $h_t^4 = f_{t-8}$. This setup enables our algorithm to achieve a sharp D-DGap bound of $\widetilde{O}(1)$ in stationary environments or periodic scenarios with cycles of 2, 3, 4, 5, 6, 7, or 8. In the Integration Module, we employ Successive Reduction of Search Space for joint updates, maintaining computational costs within acceptable limits. We also apply the doubling trick (Schapire et al., 1995) to eliminate the algorithms' dependence on the time horizon $T$.

We conduct $10^6$ rounds for each case and record the time-averaged D-DGap. The results in Figure 2 align with theoretical expectations. In Case I, both algorithms achieve near-$\widetilde{O}(1)$ D-DGaps with stable comparators, as demonstrated by their similar performance in Figure 2a. However, with highly non-stationary comparators, our algorithm maintains this bound, while Zhang et al. (2022b)'s degrades to $\widetilde{O}(\sqrt{T})$, as shown in Figure 2c, where our advantage grows over time. In Cases II and III, our algorithm consistently outperforms Zhang et al. (2022b). Notably, in Figure 2f, our algorithm converges successfully, while Zhang et al. (2022b)'s fails to converge. In Case IV, both algorithms perform comparably, as shown in Figures 2j to 2l. Our algorithm guarantees minimax optimality, while Zhang et al. (2022b)'s algorithm, despite lacking tight bounds, demonstrates empirical success due to the meta-expert framework.

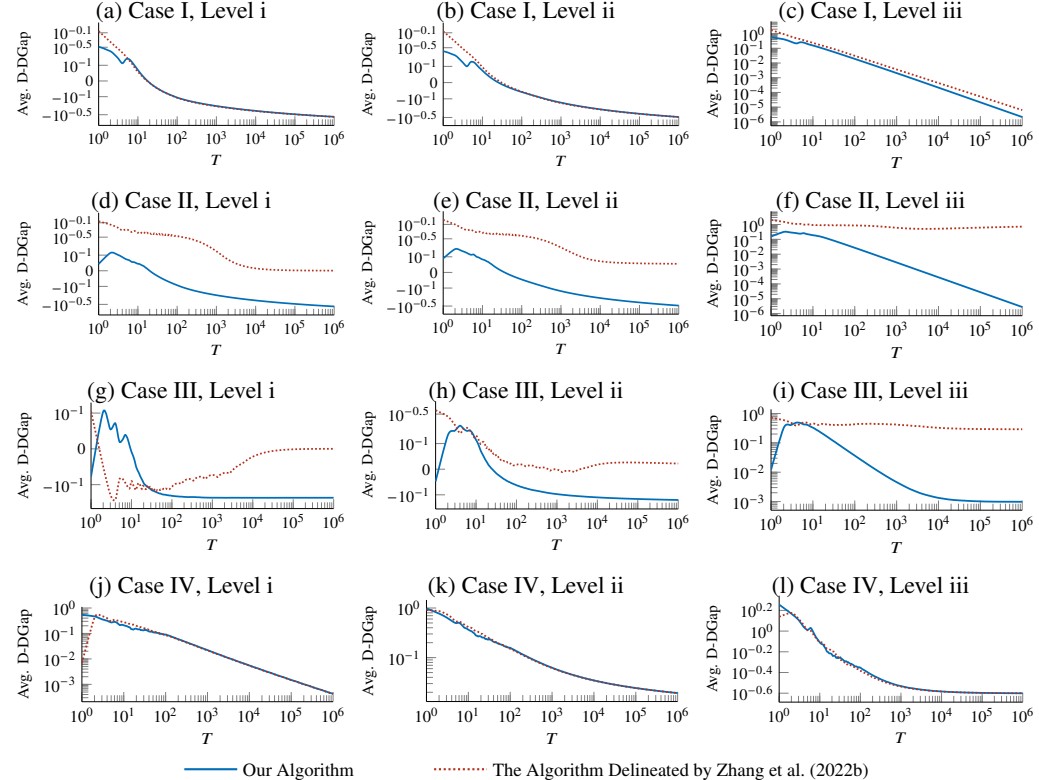

Figure 2: Time-Averaged D-DGaps of Algorithms

## 6 CONCLUSION

This paper presents the first investigation into the dynamic duality gap (D-DGap) in Online Convex-Concave Optimization (OCCO). By utilizing a modular algorithmic structure, it efficiently adapts to varying degrees of non-stationarity and leverages the most accurate predictors. The Integration Module extends the idea of meta-expert framework, guaranteeing optimal performance across varying environments.

To achieve the optimal D-DGap upper bound, our Integrated Module requires a joint optimization subroutine. This subroutine involves a two-dimensional bounded optimization problem with a continuous objective function, which is generally not considered computationally intensive. However, in online scenarios that demand rapid decision-making, the computational cost can become substantial. A promising direction for future research is to explore a trade-off between relaxing the strict guarantees on the D-DGap bound and reducing computational costs, thereby achieving a better balance between these two factors in time-sensitive applications.

### REPRODUCIBILITY

The technical appendix, located after the References, provides detailed proofs for all propositions and theorems presented in the main text. Additionally, the code for the proposed algorithm is included in the Supplementary Material to facilitate experimental validation.

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

## A   PROOF OF PROPOSITION 1

*Proof of Proposition 1.* Independently applying two ADER or ADER-like algorithms results in the following bounds:

$$\sum_{t=1}^{T} \Big( f_t(\overline{x}_t, y_t) - f_t(u_t, y_t) \Big) \leq \widetilde{O}\left( \sqrt{(1 + P_T^u)\, T} \right),$$

$$\sum_{t=1}^{T} \Big( f_t(x_t, v_t) - f_t(x_t, \overline{y}_t) \Big) \leq \widetilde{O}\left( \sqrt{(1 + P_T^v)\, T} \right),$$

where $P_T^u = \sum_{t=1}^{T} \|u_t - u_{t-1}\|$ and $P_T^v = \sum_{t=1}^{T} \|v_t - v_{t-1}\|$. For specially chosen comparators $x_t' = \arg\min_{x \in X} f_t(x, y_t)$ and $y_t' = \arg\max_{y \in Y} f_t(x_t, y)$, we also have:

$$\sum_{t=1}^{T} \Big( f_t(\overline{x}_t, y_t) - f_t(u_t, y_t) \Big) \leq \sum_{t=1}^{T} \Big( f_t(\overline{x}_t, y_t) - f_t(x_t', y_t) \Big) \leq \widetilde{O}\left( \sqrt{(1 + C_T^x)\, T} \right),$$

$$\sum_{t=1}^{T} \Big( f_t(x_t, v_t) - f_t(x_t, \overline{y}_t) \Big) \leq \sum_{t=1}^{T} \Big( f_t(x_t, y_t') - f_t(x_t, \overline{y}_t) \Big) \leq \widetilde{O}\left( \sqrt{(1 + C_T^y)\, T} \right),$$

where $C_T^x = \sum_{t=1}^{T} \|x_t' - x_{t-1}'\|$ and $C_T^y = \sum_{t=1}^{T} \|y_t' - y_{t-1}'\|$.

The desired result follows by combining these inequalities. □

## B   PROOF OF PROPOSITION 2

*Proof of Proposition 2.* Let $\mathscr{F}$ denote all convex-concave functions satisfying Assumption 2, and let $\mathscr{L}_X(G) = \{ \ell \text{ is convex} \mid \sup_{x \in X} \|\partial \ell(x)\| \leq G \}$. The key to the proof is to convert OCCO into a pair of OCO problems:

$$\sup_{f_1, \cdots, f_T \in \mathscr{F}} \left( \sup_{P_T \leq P} \Big( f_t(x_t, v_t) - f_t(u_t, y_t) \Big) \right)$$

$$\geq \sup_{f_t(x,y) = \alpha_t(x) - \beta_t(y) \in \mathscr{F},\, t \in \{1, \cdots, T\}} \left( \max_{P_T^u \leq p,\, P_T^v \leq P-p} \sum_{t=1}^{T} \Big( f_t(x_t, v_t) - f_t(u_t, y_t) \Big) \right)$$

$$= \sup_{\alpha_1, \cdots, \alpha_T \in \mathscr{L}_X(G_X)} \left( \max_{P_T^u \leq p} \sum_{t=1}^{T} \Big( \alpha_t(x_t) - \alpha_t(u_t) \Big) \right) \tag{6a}$$

$$+ \sup_{\beta_1, \cdots, \beta_T \in \mathscr{L}_Y(G_Y)} \left( \max_{P_T^v \leq P-p} \sum_{t=1}^{T} \Big( \beta_t(y_t) - \beta_t(v_t) \Big) \right), \tag{6b}$$

where the first "$\geq$" results from the consideration of the special form of $f_t$ as $f_t(x, y) = \alpha_t(x) - \beta_t(y)$, $\alpha_t$ and $\beta_t$ are both convex functions, $P_T^u = \sum_{t=1}^{T} \|u_t - u_{t-1}\|$, and $P_T^v = \sum_{t=1}^{T} \|v_t - v_{t-1}\|$. Note that

$$\text{Equation (6a)} \geq \min \left\{ \begin{array}{c} \displaystyle \sup_{\alpha_1, \cdots, \alpha_T \in \mathscr{L}_X(G_X)} \left( \max_{P_T^u \leq p} \sum_{t=1}^{T} \Big( \alpha_t(x_t) - \alpha_t(u_t) \Big) \right), \\[2em] \displaystyle \sup_{\alpha_1, \cdots, \alpha_T \in \mathscr{L}_X(G_X)} \left( \max_{P_T^u \leq C_T^x} \sum_{t=1}^{T} \Big( \alpha_t(x_t) - \alpha_t(u_t) \Big) \right) \end{array} \right\}$$

$$\geq \Omega\left( \min \left\{ \sqrt{(1 + p)T},\ \sqrt{(1 + C_T^x)T} \right\} \right),$$

where $C_T^x = \sum_{t=1}^{T} \|x_t' - x_{t-1}'\|$, $x_t' = \arg\min_{x \in X} f_t(x, y_t)$, and the second "$\geq$" is derived from Theorem 2 in Zhang et al. (2018), which establishes the lower bound on regret for OCO. Likewise,

$$\text{Equation (6b)} \geq \Omega\left( \min \left\{ \sqrt{(1 + P - p)T},\ \sqrt{(1 + C_T^y)T} \right\} \right),$$

where $C_T^y = \sum_{t=1}^{T} \|y_t' - y_{t-1}'\|$, $y_t' = \arg\max_{y \in Y} f_t(x_t, y)$. In conclusion, we have that

$$\sup_{f_1, \cdots, f_T \in \mathscr{F}} \left( \sup_{P_T \leq P} \Big( f_t(x_t, v_t) - f_t(u_t, y_t) \Big) \right) \geq \Omega \left( \min \left\{ \sqrt{(1+P)T}, \sqrt{(1+C_T)T} \right\} \right). \quad \square$$

## C   PROOF OF THEOREM 3

Theorem 8 presents the comprehensive version of Theorem 3, offering detailed specifications for the adaptive learning rates, in accordance with the methodology described by Campolongo & Orabona (2021).

**Theorem 8** (Performance Guarantee for the First Expert). *Under Assumptions 1 and 2. If the learning rates satisfy the following equations:*

$$\eta_t = L_\phi D_X (T+1) / \big( \epsilon + \textstyle\sum_{\tau=1}^{t-1} \delta_\tau^x \big), \qquad \gamma_t = L_\psi D_Y (T+1) / \big( \epsilon + \textstyle\sum_{\tau=1}^{t-1} \delta_\tau^y \big),$$

$$\delta_t^x = \big[ f_t(\widehat{x}_t, \widehat{y}_t),\; f_t(\widehat{x}_t, \overline{y}_t) \big] \boldsymbol{\omega}_t - \big[ h_t(\widehat{x}_t, \widehat{y}_t),\; h_t(\widehat{x}_t, \overline{y}_t) \big] \boldsymbol{\omega}_t$$
$$+ \big[ h_t(\widetilde{x}_{t+1}, \widehat{y}_t),\; h_t(\widetilde{x}_{t+1}, \overline{y}_t) \big] \boldsymbol{\omega}_t - \big[ f_t(\widetilde{x}_{t+1}, \widehat{y}_t),\; f_t(\widetilde{x}_{t+1}, \overline{y}_t) \big] \boldsymbol{\omega}_t,$$

$$\delta_t^y = \big[ h_t(\widehat{x}_t, \widehat{y}_t),\; h_t(\overline{x}_t, \widehat{y}_t) \big] \boldsymbol{w}_t - \big[ f_t(\widehat{x}_t, \widehat{y}_t),\; f_t(\overline{x}_t, \widehat{y}_t) \big] \boldsymbol{w}_t$$
$$+ \big[ f_t(\widehat{x}_t, \widetilde{y}_{t+1}),\; f_t(\overline{x}_t, \widetilde{y}_{t+1}) \big] \boldsymbol{w}_t - \big[ h_t(\widehat{x}_t, \widetilde{y}_{t+1}),\; h_t(\overline{x}_t, \widetilde{y}_{t+1}) \big] \boldsymbol{w}_t,$$

*where $\epsilon > 0$ prevents initial learning rates from being infinite. Then the following inequality holds:*

$$\sum_{t=1}^{T} \Big( f_t(x_t, v_t) - \boldsymbol{w}_t^{\mathrm{T}} \boldsymbol{A}_t^{:,1} \Big) + \sum_{t=1}^{T} \Big( \boldsymbol{A}_t^{1,:} \boldsymbol{\omega}_t - f_t(u_t, y_t) \Big) \leq 2\epsilon + 8 \sum_{t=1}^{T} \rho(f_t, h_t),$$

*where $\rho(f_t, h_t) = \max_{x \in X, y \in Y} |f_t(x, y) - h_t(x, y)|$ measures the distance between $f_t$ and $h_t$.*

*Proof.* The updates for the first expert can be rearranged as follows:

$$(\widehat{x}_t, \widehat{y}_t) = \arg\min_{x \in X} \max_{y \in Y} \boldsymbol{w}_t^{\mathrm{T}} \begin{bmatrix} h_t(x, y), & h_t(x, \overline{y}_t) \\ h_t(\overline{x}_t, y), & h_t(\overline{x}_t, \overline{y}_t) \end{bmatrix} \boldsymbol{\omega}_t + \frac{w_t^1}{\eta_t} B_\phi\big(x, \widetilde{x}_t^\phi\big) - \frac{\omega_t^1}{\gamma_t} B_\psi\big(y, \widetilde{y}_t^\psi\big),$$

$$\widetilde{x}_{t+1} = \arg\min_{x \in X} \eta_t \big[ f_t(x, \widehat{y}_t),\; f_t(x, \overline{y}_t) \big] \boldsymbol{\omega}_t + B_\phi\big(x, \widetilde{x}_t^\phi\big), \tag{7}$$

$$\widetilde{y}_{t+1} = \arg\max_{y \in Y} \gamma_t \big[ f_t(\widehat{x}_t, y),\; f_t(\overline{x}_t, y) \big] \boldsymbol{w}_t - B_\psi\big(y, \widetilde{y}_t^\psi\big),$$

Let's derive the upper bound for the right-hand side of the metric. Note that

$$\begin{aligned}
\boldsymbol{A}_t^{1,:} \boldsymbol{\omega}_t - f_t(u_t, y_t) \leq\; & \big[ f_t(\widehat{x}_t, \widehat{y}_t),\; f_t(\widehat{x}_t, \overline{y}_t) \big] \boldsymbol{\omega}_t - \big[ h_t(\widehat{x}_t, \widehat{y}_t),\; h_t(\widehat{x}_t, \overline{y}_t) \big] \boldsymbol{\omega}_t \\
& + \big[ h_t(\widehat{x}_t, \widehat{y}_t),\; h_t(\widehat{x}_t, \overline{y}_t) \big] \boldsymbol{\omega}_t - \big[ h_t(\widetilde{x}_{t+1}, \widehat{y}_t),\; h_t(\widetilde{x}_{t+1}, \overline{y}_t) \big] \boldsymbol{\omega}_t \\
& + \big[ h_t(\widetilde{x}_{t+1}, \widehat{y}_t),\; h_t(\widetilde{x}_{t+1}, \overline{y}_t) \big] \boldsymbol{\omega}_t - \big[ f_t(\widetilde{x}_{t+1}, \widehat{y}_t),\; f_t(\widetilde{x}_{t+1}, \overline{y}_t) \big] \boldsymbol{\omega}_t \\
& + \big[ f_t(\widetilde{x}_{t+1}, \widehat{y}_t),\; f_t(\widetilde{x}_{t+1}, \overline{y}_t) \big] \boldsymbol{\omega}_t - \big[ f_t(u_t, \widehat{y}_t),\; f_t(u_t, \overline{y}_t) \big] \boldsymbol{\omega}_t.
\end{aligned}$$

By applying the first-order optimality condition to Equation (7), we obtain

$$\big\langle \eta_t \big[ \nabla_x h_t(\widehat{x}_t, \widehat{y}_t),\; \nabla_x h_t(\widehat{x}_t, \overline{y}_t) \big] \boldsymbol{\omega}_t + \widehat{x}_t^\phi - \widetilde{x}_t^\phi,\; \widehat{x}_t - x' \big\rangle \leq 0, \qquad \forall x' \in X,$$

$$\big\langle \eta_t \big[ \nabla_x f_t(\widetilde{x}_{t+1}, \widehat{y}_t),\; \nabla_x f_t(\widetilde{x}_{t+1}, \overline{y}_t) \big] \boldsymbol{\omega}_t + \widetilde{x}_{t+1}^\phi - \widetilde{x}_t^\phi,\; \widetilde{x}_{t+1} - x' \big\rangle \leq 0, \qquad \forall x' \in X,$$

which implies that

$$\begin{aligned}
\big[ h_t(\widehat{x}_t, \widehat{y}_t),\; & h_t(\widehat{x}_t, \overline{y}_t) \big] \boldsymbol{\omega}_t - \big[ h_t(\widetilde{x}_{t+1}, \widehat{y}_t),\; h_t(\widetilde{x}_{t+1}, \overline{y}_t) \big] \boldsymbol{\omega}_t \\
& \leq \big\langle \big[ \nabla_x h_t(\widehat{x}_t, \widehat{y}_t),\; \nabla_x h_t(\widehat{x}_t, \overline{y}_t) \big] \boldsymbol{\omega}_t,\; \widehat{x}_t - \widetilde{x}_{t+1} \big\rangle \\
& \leq \big\langle \widetilde{x}_t^\phi - \widehat{x}_t^\phi,\; \widehat{x}_t - \widetilde{x}_{t+1} \big\rangle / \eta_t \\
& = \big( B_\phi\big(\widetilde{x}_{t+1}, \widetilde{x}_t^\phi\big) - B_\phi\big(\widetilde{x}_{t+1}, \widehat{x}_t^\phi\big) - B_\phi\big(\widehat{x}, \widetilde{x}_t^\phi\big) \big) / \eta_t,
\end{aligned} \tag{8a}$$

$$\begin{aligned}
\big[ f_t(\widetilde{x}_{t+1}, \widehat{y}_t),\; & f_t(\widetilde{x}_{t+1}, \overline{y}_t) \big] \boldsymbol{\omega}_t - \big[ f_t(u_t, \widehat{y}_t),\; f_t(u_t, \overline{y}_t) \big] \boldsymbol{\omega}_t \\
& \leq \big\langle \big[ \nabla_x f_t(\widetilde{x}_{t+1}, \widehat{y}_t),\; \nabla_x f_t(\widetilde{x}_{t+1}, \overline{y}_t) \big] \boldsymbol{\omega}_t,\; \widetilde{x}_{t+1} - u_t \big\rangle \\
& \leq \big\langle \widetilde{x}_t^\phi - \widetilde{x}_{t+1}^\phi,\; \widetilde{x}_{t+1} - u_t \big\rangle / \eta_t \\
& = \big( B_\phi\big(u_t, \widetilde{x}_t^\phi\big) - B_\phi\big(u_t, \widetilde{x}_{t+1}^\phi\big) - B_\phi\big(\widetilde{x}_{t+1}, \widetilde{x}_t^\phi\big) \big) / \eta_t.
\end{aligned} \tag{8b}$$

Now we have that

$$
\begin{aligned}
\boldsymbol{A}_t^{1,:}\,\boldsymbol{\omega}_t - f_t(u_t, y_t) &\leq \big[f_t(\widehat{x}_t, \widehat{y}_t),\ f_t(\widehat{x}_t, \overline{y}_t)\big]\boldsymbol{\omega}_t - \big[h_t(\widehat{x}_t, \widehat{y}_t),\ h_t(\widehat{x}_t, \overline{y}_t)\big]\boldsymbol{\omega}_t \\
&\quad + \big[h_t(\widetilde{x}_{t+1}, \widehat{y}_t),\ h_t(\widetilde{x}_{t+1}, \overline{y}_t)\big]\boldsymbol{\omega}_t - \big[f_t(\widetilde{x}_{t+1}, \widehat{y}_t),\ f_t(\widetilde{x}_{t+1}, \overline{y}_t)\big]\boldsymbol{\omega}_t \\
&\quad + \big(B_\phi\big(u_t, \widetilde{x}_t^\phi\big) - B_\phi\big(u_t, \widetilde{x}_{t+1}^\phi\big)\big)/\eta_t \\
&= \big(B_\phi\big(u_t, \widetilde{x}_t^\phi\big) - B_\phi\big(u_t, \widetilde{x}_{t+1}^\phi\big)\big)/\eta_t + \delta_t^x,
\end{aligned}
\tag{9}
$$

where $\delta_t^x \geq 0$. This can be obtained by adding Equation (8a) and the following inequality:

$$
\big[f_t(\widehat{x}_t, \widehat{y}_t),\ f_t(\widehat{x}_t, \overline{y}_t)\big]\boldsymbol{\omega}_t + B_\phi\big(\widehat{x}_t, \widetilde{x}_t^\phi\big)/\eta_t \geq \big[f_t(\widetilde{x}_{t+1}, \widehat{y}_t),\ f_t(\widetilde{x}_{t+1}, \overline{y}_t)\big]\boldsymbol{\omega}_t + B_\phi\big(\widetilde{x}_{t+1}, \widetilde{x}_t^\phi\big)/\eta_t,
$$

which corresponds to the optimality of $\widetilde{x}_{t+1}$.

Summing Equation (9) over time yields

$$
\sum_{t=1}^{T}\Big(\boldsymbol{A}_t^{1,:}\,\boldsymbol{\omega}_t - f_t(u_t, y_t)\Big) \leq \sum_{t=1}^{T}\frac{1}{\eta_t}\Big(B_\phi\big(u_t, \widetilde{x}_t^\phi\big) - B_\phi\big(u_t, \widetilde{x}_{t+1}^\phi\big)\Big) + \sum_{t=1}^{T}\delta_t^x,
$$

Due to the non-increasing nature of the learning rate $\eta_t$, $B_\phi$ is upper bounded by $L_\phi D_X$ and is $L_\phi$-Lipschitz with respect to its first variable. Therefore, we have that

$$
\begin{aligned}
&\sum_{t=1}^{T}\frac{1}{\eta_t}\Big(B_\phi\big(u_t, \widetilde{x}_t^\phi\big) - B_\phi\big(u_t, \widetilde{x}_{t+1}^\phi\big)\Big) \\
&\leq \sum_{t=1}^{T}\frac{1}{\eta_t}\Big(B_\phi\big(u_t, \widetilde{x}_t^\phi\big) - B_\phi\big(u_{t-1}, \widetilde{x}_t^\phi\big)\Big) + \frac{B_\phi\big(u_0, \widetilde{x}_1^\phi\big)}{\eta_1} + \sum_{t=2}^{T}\left(\frac{1}{\eta_t} - \frac{1}{\eta_{t-1}}\right)B_\phi\big(u_{t-1}, \widetilde{x}_t^\phi\big) \\
&\leq \frac{L_\phi D_X}{\eta_T} + \sum_{t=1}^{T}\frac{L_\phi}{\eta_t}\,\|u_t - u_{t-1}\|.
\end{aligned}
$$

Applying the prescribed learning rate yields

$$
\sum_{t=1}^{T}\Big(\boldsymbol{A}_t^{1,:}\,\boldsymbol{\omega}_t - f_t(u_t, y_t)\Big) \leq \frac{L_\phi}{\eta_T}\left(D_X + P_T^u\right) + \sum_{t=1}^{T}\delta_t^x \leq \epsilon + 2\sum_{t=1}^{T}\delta_t^x,
$$

where $P_T^u = \sum_{t=1}^{T}\|u_t - u_{t-1}\| \leq D_X T$. Note that

$$
\begin{aligned}
\delta_t^x &= \big[f_t(\widehat{x}_t, \widehat{y}_t),\ f_t(\widehat{x}_t, \overline{y}_t)\big]\boldsymbol{\omega}_t - \big[h_t(\widehat{x}_t, \widehat{y}_t),\ h_t(\widehat{x}_t, \overline{y}_t)\big]\boldsymbol{\omega}_t \\
&\quad + \big[h_t(\widetilde{x}_{t+1}, \widehat{y}_t),\ h_t(\widetilde{x}_{t+1}, \overline{y}_t)\big]\boldsymbol{\omega}_t - \big[f_t(\widetilde{x}_{t+1}, \widehat{y}_t),\ f_t(\widetilde{x}_{t+1}, \overline{y}_t)\big]\boldsymbol{\omega}_t \\
&\leq 2\max_{x\in\{\widehat{x}_t, \overline{x}_t, \widetilde{x}_{t+1}\}, y\in\{\widehat{y}_t, \overline{y}_t\}}|f_t(x, y) - h_t(x, y)| \\
&\leq 2\rho\big(f_t, h_t\big),
\end{aligned}
\tag{10}
$$

So we have that

$$
\sum_{t=1}^{T}\Big(\boldsymbol{A}_t^{1,:}\,\boldsymbol{\omega}_t - f_t(u_t, y_t)\Big) \leq \epsilon + 4\sum_{t=1}^{T}\rho(f_t, h_t).
$$

Likewise, the upper bound for the left-hand side of the metric is as follows:

$$
\sum_{t=1}^{T}\Big(f_t(x_t, v_t) - \boldsymbol{w}_t^{\mathrm{T}}\boldsymbol{A}_t^{:,1}\Big) \leq \epsilon + 4\sum_{t=1}^{T}\rho(f_t, h_t).
$$

Adding the above two inequalities yields the desired conclusion. □

## D  PROOF OF THEOREM 4

Theorem 9 presents the full version of Theorem 4, offering detailed settings for the adaptive learning rates, in accordance with the methodology described by Campolongo & Orabona (2021).

**Theorem 9** (Performance Guarantee for the Meta Layer). *Under Assumption 2, and assume that $T \geq 2$. If the learning rates satisfy the following inequalities:*

$$\theta_t = (\ln T)/\big(\epsilon + \textstyle\sum_{\tau=1}^{t-1} \Delta_\tau^x\big), \quad \Delta_t^x = (\boldsymbol{w}_t - \widetilde{\boldsymbol{w}}_{t+1})^{\mathrm{T}} (\boldsymbol{A}_t - \boldsymbol{\Lambda}_t) \boldsymbol{\omega}_t - \mathrm{KL}(\widetilde{\boldsymbol{w}}_{t+1}, \boldsymbol{w}_t)/\theta_t,$$

$$\vartheta_t = (\ln T)/\big(\epsilon + \textstyle\sum_{\tau=1}^{t-1} \Delta_\tau^y\big), \quad \Delta_t^y = -\boldsymbol{w}_t^{\mathrm{T}} (\boldsymbol{A}_t - \boldsymbol{\Lambda}_t) (\boldsymbol{\omega}_t - \widetilde{\boldsymbol{\omega}}_{t+1}) - \mathrm{KL}(\widetilde{\boldsymbol{\omega}}_{t+1}, \boldsymbol{\omega}_t)/\vartheta_t,$$

*where $\epsilon > 0$ prevents initial learning rates from being infinite. Then the meta layer of the Integration Module enjoys the following inequality: $\forall \boldsymbol{u}, \boldsymbol{v} \in \triangle_2$,*

$$\sum_{t=1}^{T} \big(\boldsymbol{w}_t^{\mathrm{T}} \boldsymbol{A}_t \boldsymbol{v} - \boldsymbol{u}^{\mathrm{T}} \boldsymbol{A}_t \boldsymbol{\omega}_t\big) \leq O(1) + 4 \min \left\{ 2 \sum_{t=1}^{T} \rho(f_t, h_t), \sqrt{(4 + \ln T) \textstyle\sum_{t=1}^{T} \rho^2(f_t, h_t)} \right\}.$$

*Proof.* The meta layer updates of the Integration Module can be rearranged as follows:

$$(\boldsymbol{w}_t, \boldsymbol{\omega}_t) = \arg\min_{\boldsymbol{w} \in \triangle_2^\alpha} \max_{\boldsymbol{\omega} \in \triangle_2^\alpha} \boldsymbol{w}^{\mathrm{T}} \boldsymbol{\Lambda}_t \boldsymbol{\omega} + \mathrm{KL}(\boldsymbol{w}, \widetilde{\boldsymbol{w}}_t)/\theta_t - \mathrm{KL}(\boldsymbol{\omega}, \widetilde{\boldsymbol{\omega}}_t)/\vartheta_t,$$

$$\widetilde{\boldsymbol{w}}_{t+1} = \arg\min_{\boldsymbol{w} \in \triangle_2^\alpha} \langle \theta_t \boldsymbol{A}_t \boldsymbol{\omega}_t, \boldsymbol{w} \rangle + \mathrm{KL}(\boldsymbol{w}, \widetilde{\boldsymbol{w}}_t), \tag{11}$$

$$\widetilde{\boldsymbol{\omega}}_{t+1} = \arg\max_{\boldsymbol{\omega} \in \triangle_2^\alpha} \langle \vartheta_t \boldsymbol{A}_t^{\mathrm{T}} \boldsymbol{w}_t, \boldsymbol{\omega} \rangle - \mathrm{KL}(\boldsymbol{\omega}, \widetilde{\boldsymbol{\omega}}_t),$$

where $\alpha = 2/T$. Let $\boldsymbol{1} = [1, 1]^{\mathrm{T}}$. By inserting an auxiliary term $\boldsymbol{w} = \alpha \boldsymbol{1}/2 + (1 - \alpha) \boldsymbol{u}$, we get

$$\sum_{t=1}^{T} (\boldsymbol{w}_t - \boldsymbol{u})^{\mathrm{T}} \boldsymbol{A}_t \boldsymbol{\omega}_t = \sum_{t=1}^{T} (\boldsymbol{w}_t - \boldsymbol{w})^{\mathrm{T}} \boldsymbol{A}_t \boldsymbol{\omega}_t + \sum_{t=1}^{T} (\boldsymbol{w} - \boldsymbol{u})^{\mathrm{T}} \boldsymbol{A}_t \boldsymbol{\omega}_t, \qquad \forall \boldsymbol{u} \in \triangle_2,$$

where

$$\sum_{t=1}^{T} (\boldsymbol{w} - \boldsymbol{u})^{\mathrm{T}} \boldsymbol{A}_t \boldsymbol{\omega}_t \leq T \left\| \frac{\alpha}{2} \boldsymbol{1} - \alpha \boldsymbol{u} \right\|_1 \|\boldsymbol{A}_t\|_\infty \leq 2\alpha T M = 4M.$$

Let $\boldsymbol{\ell}_t = \boldsymbol{A}_t \boldsymbol{\omega}_t$ and $\boldsymbol{\hbar}_t = \boldsymbol{\Lambda}_t \boldsymbol{\omega}_t$. Applying the first-order optimality condition to Equation (11) yields

$$(\boldsymbol{w}_t - \boldsymbol{w}')^{\mathrm{T}} (\theta_t \boldsymbol{\hbar}_t + \ln \boldsymbol{w}_t + \boldsymbol{1} - \ln \widetilde{\boldsymbol{w}}_t) \leq 0, \qquad \forall \boldsymbol{w}' \in \triangle_2^\alpha,$$

$$(\widetilde{\boldsymbol{w}}_{t+1} - \boldsymbol{w}')^{\mathrm{T}} (\theta_t \boldsymbol{\ell}_t + \ln \widetilde{\boldsymbol{w}}_{t+1} + \boldsymbol{1} - \ln \widetilde{\boldsymbol{w}}_t) \leq 0, \qquad \forall \boldsymbol{w}' \in \triangle_2^\alpha.$$

Thus we have that

$$(\boldsymbol{w}_t - \boldsymbol{w})^{\mathrm{T}} \boldsymbol{A}_t \boldsymbol{\omega}_t$$

$$= (\boldsymbol{w}_t - \widetilde{\boldsymbol{w}}_{t+1})^{\mathrm{T}} (\boldsymbol{\ell}_t - \boldsymbol{\hbar}_t) + (\boldsymbol{w}_t - \widetilde{\boldsymbol{w}}_{t+1})^{\mathrm{T}} \boldsymbol{\hbar}_t + (\widetilde{\boldsymbol{w}}_{t+1} - \boldsymbol{w})^{\mathrm{T}} \boldsymbol{\ell}_t$$

$$\leq (\boldsymbol{w}_t - \widetilde{\boldsymbol{w}}_{t+1})^{\mathrm{T}} (\boldsymbol{\ell}_t - \boldsymbol{\hbar}_t) + \frac{1}{\theta_t}(\boldsymbol{w}_t - \widetilde{\boldsymbol{w}}_{t+1})^{\mathrm{T}} \ln \frac{\widetilde{\boldsymbol{w}}_t}{\boldsymbol{w}_t} + \frac{1}{\theta_t}(\widetilde{\boldsymbol{w}}_{t+1} - \boldsymbol{w})^{\mathrm{T}} \ln \frac{\widetilde{\boldsymbol{w}}_t}{\widetilde{\boldsymbol{w}}_{t+1}}$$

$$\leq (\boldsymbol{w}_t - \widetilde{\boldsymbol{w}}_{t+1})^{\mathrm{T}} (\boldsymbol{\ell}_t - \boldsymbol{\hbar}_t) - \frac{1}{\theta_t}\mathrm{KL}(\widetilde{\boldsymbol{w}}_{t+1}, \boldsymbol{w}_t) - \frac{1}{\theta_t}\mathrm{KL}(\boldsymbol{w}_t, \widetilde{\boldsymbol{w}}_t) + \frac{1}{\theta_t}\boldsymbol{w}^{\mathrm{T}} \ln \frac{\widetilde{\boldsymbol{w}}_{t+1}}{\widetilde{\boldsymbol{w}}_t}$$

$$\leq \Delta_t^x + \boldsymbol{w}^{\mathrm{T}} (\ln \widetilde{\boldsymbol{w}}_{t+1} - \ln \widetilde{\boldsymbol{w}}_t)/\theta_t.$$

Summing over time yields

$$\sum_{t=1}^{T} (\boldsymbol{w}_t - \boldsymbol{w})^{\mathrm{T}} \boldsymbol{A}_t \boldsymbol{\omega}_t \leq \sum_{t=1}^{T} \Delta_t^x + \sum_{t=1}^{T} \frac{1}{\theta_t}\Big(\mathrm{KL}(\boldsymbol{w}, \widetilde{\boldsymbol{w}}_t) - \mathrm{KL}(\boldsymbol{w}, \widetilde{\boldsymbol{w}}_{t+1})\Big)$$

$$= \sum_{t=1}^{T} \Delta_t^x + \frac{\mathrm{KL}(\boldsymbol{w}, \widetilde{\boldsymbol{w}}_1)}{\theta_1} + \sum_{t=2}^{T} \left(\frac{1}{\theta_t} - \frac{1}{\theta_{t-1}}\right) \mathrm{KL}(\boldsymbol{w}, \widetilde{\boldsymbol{w}}_t) - \frac{\mathrm{KL}(\boldsymbol{w}, \widetilde{\boldsymbol{w}}_{T+1})}{\theta_T}$$

$$\leq \sum_{t=1}^{T} \Delta_t^x + \frac{\ln T}{\theta_1} + \sum_{t=2}^{T} \left(\frac{1}{\theta_t} - \frac{1}{\theta_{t-1}}\right) \ln T = \frac{\ln T}{\theta_T} + \sum_{t=1}^{T} \Delta_t^x,$$

where the last "$\leq$" follows from the two facts that $0 \leq \mathrm{KL}(\boldsymbol{a}, \boldsymbol{b}) \leq \ln T$ for all $\boldsymbol{a}, \boldsymbol{b} \in \triangle_2^\alpha$, and the learning rate is non-increasing. The derivation for the first fact is as follows:

$$0 \leq \mathrm{KL}(\boldsymbol{a}, \boldsymbol{b}) = \boldsymbol{a}^{\mathrm{T}} \ln \frac{\boldsymbol{a}}{\boldsymbol{b}} \leq \ln \boldsymbol{a}^{\mathrm{T}} \frac{\boldsymbol{a}}{\boldsymbol{b}} \leq \ln \left\| \frac{\boldsymbol{a}}{\boldsymbol{b}} \right\|_\infty \leq \ln T.$$

The reason for the second fact is $\Delta_t^x \geq 0$, which can be obtained by adding the following two inequalities:

$$\boldsymbol{w}_t^{\mathrm{T}}\boldsymbol{\ell}_t + \mathrm{KL}(\boldsymbol{w}_t, \widetilde{\boldsymbol{w}}_t)/\theta_t \geq \widetilde{\boldsymbol{w}}_{t+1}^{\mathrm{T}}\boldsymbol{\ell}_t + \mathrm{KL}(\widetilde{\boldsymbol{w}}_{t+1}, \widetilde{\boldsymbol{w}}_t)/\theta_t, \qquad \text{// The optimality of } \widetilde{\boldsymbol{w}}_{t+1},$$

$$(\boldsymbol{w}_t - \widetilde{\boldsymbol{w}}_{t+1})^{\mathrm{T}}(\theta_t \boldsymbol{\hbar}_t + \ln \boldsymbol{w}_t - \ln \widetilde{\boldsymbol{w}}_t) \leq 0, \qquad \text{// First-order optimality condition.}$$

Now applying the prescribed learning rate yields

$$\sum_{t=1}^{T}(\boldsymbol{w}_t - \boldsymbol{w})^{\mathrm{T}}\boldsymbol{A}_t\boldsymbol{\omega}_t \leq \epsilon + 2\sum_{t=1}^{T}\Delta_t^x.$$

Next, we estimate the upper bound of the above inequality. Note that

$$\Delta_t^x \leq \|\boldsymbol{w}_t - \widetilde{\boldsymbol{w}}_{t+1}\|_1 \|\boldsymbol{A}_t - \boldsymbol{\Lambda}_t\|_\infty - \frac{1}{2\theta_t}\|\boldsymbol{w}_t - \widetilde{\boldsymbol{w}}_{t+1}\|_1^2$$

$$\leq \min\left\{2\|\boldsymbol{A}_t - \boldsymbol{\Lambda}_t\|_\infty, \frac{\theta_t}{2}\|\boldsymbol{A}_t - \boldsymbol{\Lambda}_t\|_\infty^2\right\}$$

Thus we have that

$$\left(\sum_{t=1}^{T}\Delta_t^x\right)^2 = \sum_{t=1}^{T}(\Delta_t^x)^2 + 2\sum_{t=1}^{T}\Delta_t^x\sum_{\tau=1}^{t-1}\Delta_\tau^x \leq \sum_{t=1}^{T}(\Delta_t^x)^2 + 2\sum_{t=1}^{T}\Delta_t^x\left(\frac{\ln T}{\theta_t} - \epsilon\right)$$

$$\leq \sum_{t=1}^{T}4\|\boldsymbol{A}_t - \boldsymbol{\Lambda}_t\|_\infty^2 + \sum_{t=1}^{T}\ln T\|\boldsymbol{A}_t - \boldsymbol{\Lambda}_t\|_\infty^2$$

$$= (4 + \ln T)\sum_{t=1}^{T}\|\boldsymbol{A}_t - \boldsymbol{\Lambda}_t\|_\infty^2.$$

Combining the above three inequalities yields

$$\sum_{t=1}^{T}(\boldsymbol{w}_t - \boldsymbol{w})^{\mathrm{T}}\boldsymbol{A}_t\boldsymbol{\omega}_t \leq \epsilon + 2\min\left\{2\sum_{t=1}^{T}\|\boldsymbol{A}_t - \boldsymbol{\Lambda}_t\|_\infty, \sqrt{(4 + \ln T)\sum_{t=1}^{T}\|\boldsymbol{A}_t - \boldsymbol{\Lambda}_t\|_\infty^2}\right\}.$$

In conclusion, $\forall \boldsymbol{u} \in \triangle_2$:

$$\sum_{t=1}^{T}(\boldsymbol{w}_t - \boldsymbol{u})^{\mathrm{T}}\boldsymbol{A}_t\boldsymbol{\omega}_t = \sum_{t=1}^{T}(\boldsymbol{w}_t - \boldsymbol{w})^{\mathrm{T}}\boldsymbol{A}_t\boldsymbol{\omega}_t + \sum_{t=1}^{T}(\boldsymbol{w} - \boldsymbol{u})^{\mathrm{T}}\boldsymbol{A}_t\boldsymbol{\omega}_t$$

$$\leq \epsilon + 2\min\left\{2\sum_{t=1}^{T}\rho(f_t, h_t), \sqrt{(4 + \ln T)\sum_{t=1}^{T}\rho^2(f_t, h_t)}\right\} + 4M,$$

where the inequality holds since

$$\|\boldsymbol{A}_t - \boldsymbol{\Lambda}_t\|_\infty = \max_{x \in \{\widehat{x}_t, \overline{x}_t\}, y \in \{\widehat{y}_t, \overline{y}_t\}}|f_t(x, y) - h_t(x, y)| \leq \rho(f_t, h_t). \tag{12}$$

Likewise, $\forall \boldsymbol{v} \in \triangle_2$:

$$\sum_{t=1}^{T}\boldsymbol{w}_t^{\mathrm{T}}\boldsymbol{A}_t(\boldsymbol{v} - \boldsymbol{\omega}_t) \leq \epsilon + 2\min\left\{2\sum_{t=1}^{T}\rho(f_t, h_t), \sqrt{(4 + \ln T)\sum_{t=1}^{T}\rho^2(f_t, h_t)}\right\} + 4M.$$

Adding the above two individual regrets yields

$$\sum_{t=1}^{T}\left(\boldsymbol{w}_t^{\mathrm{T}}\boldsymbol{A}_t\boldsymbol{v} - \boldsymbol{u}^{\mathrm{T}}\boldsymbol{A}_t\boldsymbol{\omega}_t\right) \leq O(1) + 4\min\left\{2\sum_{t=1}^{T}\rho(f_t, h_t), \sqrt{(4 + \ln T)\sum_{t=1}^{T}\rho^2(f_t, h_t)}\right\},$$

which meets the stated bound. $\qquad\square$

## E    PROOF OF THEOREM 6

*Proof of Theorem 6.* We begin by proving that the saddle point map $\mathscr{H}_t \colon (\boldsymbol{w}, \boldsymbol{\omega}) \mapsto (\widehat{x}, \widehat{y})$, where $(\widehat{x}, \widehat{y})$ is the saddle point of $H_t(\,\cdot\,,\,\cdot\,;\boldsymbol{w}, \boldsymbol{\omega})$, is well-defined and continuous.

By Sion's Minimax Theorem (Sion, 1958), since $H_t(\,\cdot\,,\,\cdot\,;\boldsymbol{w}, \boldsymbol{\omega})$ is convex in $X$, concave in $Y$, and continuous, and given that both $X$ and $Y$ are compact and convex, it follows that for every fixed pair $(\boldsymbol{w}, \boldsymbol{\omega})$, there exists a saddle point $(\widehat{x}, \widehat{y})$. Furthermore, as $H_t(\,\cdot\,,\,\cdot\,;\boldsymbol{w}, \boldsymbol{\omega})$ is strictly convex in $X$ and strictly concave in $Y$, the saddle point $(\widehat{x}, \widehat{y})$ is unique. To substantiate this uniqueness, assume for contradiction that there exists another saddle point $(\widehat{x}', \widehat{y}')$. In such a case, the following two inequalities must hold:

$$H_t(\widehat{x}', \widehat{y}'; \boldsymbol{w}, \boldsymbol{\omega}) < H_t(\widehat{x}, \widehat{y}'; \boldsymbol{w}, \boldsymbol{\omega}) < H_t(\widehat{x}, \widehat{y}; \boldsymbol{w}, \boldsymbol{\omega}),$$

$$H_t(\widehat{x}, \widehat{y}; \boldsymbol{w}, \boldsymbol{\omega}) < H_t(\widehat{x}', \widehat{y}; \boldsymbol{w}, \boldsymbol{\omega}) < H_t(\widehat{x}', \widehat{y}'; \boldsymbol{w}, \boldsymbol{\omega}),$$

which leads to a contradiction. Therefore, the saddle point $(\widehat{x}, \widehat{y})$ is unique, and $\mathscr{H}_t$ is well-defined.

Now, assume that the sequence $(\boldsymbol{w}_n, \boldsymbol{\omega}_n)$ converges to $(\boldsymbol{w}_0, \boldsymbol{\omega}_0)$ as $n \to +\infty$. To prove that $\mathscr{H}_t$ is continuous, we must show that the corresponding sequence of saddle points $(\widehat{x}_n, \widehat{y}_n)$ converges to $(\widehat{x}_0, \widehat{y}_0)$. Suppose, for contradiction, that $(\widehat{x}_n, \widehat{y}_n) \not\to (\widehat{x}_0, \widehat{y}_0)$. Since $X \times Y$ is compact, there must exist a subsequence $(\widehat{x}_{n_k}, \widehat{y}_{n_k})$ converging to some point $(\widehat{x}_0', \widehat{y}_0') \neq (\widehat{x}_0, \widehat{y}_0)$. By the continuity of $H_t$, the saddle point inequalities are preserved along the subsequence, yielding:

$$H_t(\widehat{x}_{n_k}, y; \boldsymbol{w}_{n_k}, \boldsymbol{\omega}_{n_k}) \le H_t(\widehat{x}_{n_k}, \widehat{y}_{n_k}; \boldsymbol{w}_{n_k}, \boldsymbol{\omega}_{n_k}) \le H_t(x, \widehat{y}_{n_k}; \boldsymbol{w}_{n_k}, \boldsymbol{\omega}_{n_k}), \quad \forall x \in X, \, y \in Y.$$

Taking the limit as $k \to +\infty$, we obtain:

$$H_t(\widehat{x}_0', y; \boldsymbol{w}_0, \boldsymbol{\omega}_0) \le H_t(\widehat{x}_0', \widehat{y}_0'; \boldsymbol{w}_0, \boldsymbol{\omega}_0) \le H_t(x, \widehat{y}_0'; \boldsymbol{w}_0, \boldsymbol{\omega}_0), \quad \forall x \in X, \, y \in Y.$$

This implies that $(\widehat{x}_0', \widehat{y}_0')$ satisfies the saddle point conditions for $H_t(\,\cdot\,,\,\cdot\,;\boldsymbol{w}_0, \boldsymbol{\omega}_0)$. However, since the saddle point is unique, it follows that $(\widehat{x}_0', \widehat{y}_0') = (\widehat{x}_0, \widehat{y}_0)$. This contradicts the assumption that $(\widehat{x}_0', \widehat{y}_0') \neq (\widehat{x}_0, \widehat{y}_0)$. Therefore, $\mathscr{H}_t$ is continuous.

Similarly, the map $\mathscr{W}_t$ is also well-defined and continuous. Hence, the composition $\mathscr{W}_t \circ \mathscr{H}_t$ is a continuous map from the compact convex set $\triangle_2^\alpha \times \triangle_2^\alpha$ to itself. By Brouwer's fixed-point theorem (Brouwer, 1911; Karapınar & Agarwal, 2022), there exists a point $(\boldsymbol{w}', \boldsymbol{\omega}')$ such that $(\boldsymbol{w}', \boldsymbol{\omega}') = \mathscr{W}_t \circ \mathscr{H}_t(\boldsymbol{w}', \boldsymbol{\omega}')$, and this point corresponds to the solution of Equations (3a) and (3d).

We now turn our attention to proving the second statement. Define

$$\Phi_t \colon (\boldsymbol{w}, \boldsymbol{\omega}) \mapsto \arg \max_{\boldsymbol{v} \in \triangle_2^\alpha} W_t\left(\boldsymbol{w}, \boldsymbol{v}; \mathscr{H}_t(\boldsymbol{w}, \boldsymbol{\omega})\right),$$

$$\Psi_t \colon (\boldsymbol{w}, \boldsymbol{\omega}) \mapsto \arg \min_{\boldsymbol{u} \in \triangle_2^\alpha} W_t\left(\boldsymbol{u}, \boldsymbol{\omega}; \mathscr{H}_t(\boldsymbol{w}, \boldsymbol{\omega})\right).$$

Both $\Phi_t$ and $\Psi_t$ are well-defined and continuous, as can be established following the previous proof. Then $\mathscr{F}_t \colon (w, \omega) \mapsto (\boldsymbol{w}, \boldsymbol{\omega}) \mapsto W_t\left(\boldsymbol{w}, \Phi_t(\boldsymbol{w}, \boldsymbol{\omega}); \mathscr{H}_t(\boldsymbol{w}, \boldsymbol{\omega})\right) - W_t\left(\Psi_t(\boldsymbol{w}, \boldsymbol{\omega}), \boldsymbol{\omega}; \mathscr{H}_t(\boldsymbol{w}, \boldsymbol{\omega})\right) = \sigma$ is continuous.

Note that $\max_{\boldsymbol{v} \in \triangle_2^\alpha} W_t\left(\boldsymbol{w}, \boldsymbol{v}; \widehat{x}, \widehat{y}\right) \ge W_t\left(\boldsymbol{w}, \boldsymbol{\omega}; \widehat{x}, \widehat{y}\right) \ge \min_{\boldsymbol{u} \in \triangle_2^\alpha} W_t\left(\boldsymbol{u}, \boldsymbol{\omega}; \widehat{x}, \widehat{y}\right)$, so $\sigma \ge 0$, and equality holds if and only if $(\boldsymbol{w}, \boldsymbol{\omega})$ is the saddle point of $W_t\left(\,\cdot\,,\,\cdot\,;\widehat{x}, \widehat{y}\right)$. Therefore, combining this with the fact that $(\widehat{x}, \widehat{y})$ is the saddle point of $H_t(\,\cdot\,,\,\cdot\,;\boldsymbol{w}, \boldsymbol{\omega})$, we conclude that solving the fixed point of $\mathscr{W}_t \circ \mathscr{H}_t$ is equivalent to minimizing the continuous map $\mathscr{F}_t$ to zero. □

## F    PROOF OF THEOREM 7

Theorem 10 is the full version of Theorem 7, providing detailed settings for the adaptive learning rate, in accordance with the methodology described by Campolongo & Orabona (2021).

**Theorem 10** (D-DGap for the Integration Module Using a Multi-Predictor Aggregator). *Assume the payoff function $f_t$ and all predictors $\{h_t^1, h_t^2, \cdots, h_t^d\}$ satisfy Assumption 2. Let $T \ge d$. If the Multi-Predictor Aggregator updates its learning rate according to the following equations:*

$$\zeta_t = (\ln T)\big/\big(\epsilon + \textstyle\sum_{\tau=1}^{t-1} \Delta_\tau\big), \qquad \epsilon > 0, \qquad \Delta_t = \langle \boldsymbol{L}_t, \boldsymbol{\xi}_t - \boldsymbol{\xi}_{t+1} \rangle - \mathrm{KL}(\boldsymbol{\xi}_{t+1}, \boldsymbol{\xi}_t)/\zeta_t.$$

*Then, the D-DGap upper bound for the Integration Module can be enhanced as follows:*

$$\text{D-DGap}_T \leq \widetilde{O}\left(\min\left\{\min_{k\in\{1,2,\cdots,d\}}\sum_{t=1}^{T}\rho(f_t, h_t^k), \ \sqrt{(1+\min\{P_T, C_T\})T}\right\}\right).$$

*Proof.* In the proofs of Theorems 3 and 4, the relaxed inequalities $\delta_t^x, \delta_t^y \leq 2\rho(f_t, h_t)$ and $\|\boldsymbol{A}_t - \boldsymbol{\Lambda}_t\|_\infty \leq \rho(f_t, h_t)$ are utilized (refer to Equations (10) and (12)). However, by appropriately setting the loss vector $\boldsymbol{L}_t$, these upper bounds can be tightened further, as follows:

$$\delta_t^x, \delta_t^y, 2\|\boldsymbol{A}_t - \boldsymbol{\Lambda}_t\|_\infty \leq 2\langle\boldsymbol{L}_t, \boldsymbol{\xi}_t\rangle.$$

Applying Lemma 11, we derive:

$$\sum_{t=1}^{T}\langle\boldsymbol{L}_t, \boldsymbol{\xi}_t\rangle \leq \sum_{t=1}^{T}\langle\boldsymbol{L}_t, \boldsymbol{1}_k\rangle + 2\sqrt{2M(1+\ln T)\sum_{t=1}^{T}\langle\boldsymbol{L}_t, \boldsymbol{1}_k\rangle} + O(\ln T)$$

$$= \sum_{t=1}^{T}\langle\boldsymbol{L}_t, \boldsymbol{1}_k\rangle + O\left(\sqrt{\ln T}\right)\sqrt{\sum_{t=1}^{T}\langle\boldsymbol{L}_t, \boldsymbol{1}_k\rangle} + O(\ln T)$$

$$\leq 2\sum_{t=1}^{T}\langle\boldsymbol{L}_t, \boldsymbol{1}_k\rangle + O(\ln T) \leq 2\sum_{t=1}^{T}\rho\left(f_t, h_t^k\right) + O(\ln T), \qquad \forall k = 1, 2, \cdots, d,$$

where $\boldsymbol{1}_k$ is a $d$-dimensional one-hot vector with the $k$-th element being 1. Given the arbitrariness of $k$, it follows that:

$$\sum_{t=1}^{T}\langle\boldsymbol{L}_t, \boldsymbol{\xi}_t\rangle \leq 2\min_{k\in\{1,2,\cdots,d\}}\sum_{t=1}^{T}\rho\left(f_t, h_t^k\right) + O(\ln T).$$

Therefore, the term $\sum_{t=1}^{T}\rho(f_t, h_t)$ in the performance bounds of both the meta layer and expert layer can be replaced with $\widetilde{O}\left(\min_{k\in\{1,2,\cdots,d\}}\sum_{t=1}^{T}\rho(f_t, h_t^k)\right)$, resulting in the following D-DGap upper bound:

$$\text{D-DGap}_T \leq \widetilde{O}\left(\min\left\{\min_{k\in\{1,2,\cdots,d\}}\sum_{t=1}^{T}\rho(f_t, h_t^k), \ \sqrt{(1+\min\{P_T, C_T\})T}\right\}\right). \qquad \square$$

**Lemma 11** (Static Regret for Clipped Hedge, Static Version of Corollary B.0.1 in Campolongo & Orabona (2021)). *Let $\triangle_d^\alpha$ be a $d$-dimensional $\alpha$-clipped simplex, $T \geq d$ and $\alpha = d/T$. Assume that all bounded linear losses satisfy $\boldsymbol{L}_t \geq 0$ and $\max_{t\in 1:T}\|\boldsymbol{L}_t\|_\infty = L_\infty$. If $\boldsymbol{\xi}_t$ follows the clipped Hedge:*

$$\boldsymbol{\xi}_{t+1} = \arg\min_{\boldsymbol{\xi}\in\triangle_d^a} \zeta_t\langle\boldsymbol{L}_t, \boldsymbol{\xi}\rangle + \text{KL}(\boldsymbol{\xi}, \boldsymbol{\xi}_t),$$

*where the learning rate $\zeta_t$ is determined by the following equations:*

$$\zeta_t = (\ln T)\big/\big(\epsilon + \sum_{\tau=1}^{t-1}\Delta_\tau\big), \qquad \epsilon > 0, \qquad \Delta_t = \langle\boldsymbol{L}_t, \boldsymbol{\xi}_t - \boldsymbol{\xi}_{t+1}\rangle - \text{KL}(\boldsymbol{\xi}_{t+1}, \boldsymbol{\xi}_t)/\zeta_t.$$

*Then we have that*

$$\sum_{t=1}^{T}\langle\boldsymbol{L}_t, \boldsymbol{\xi}_t - \boldsymbol{u}\rangle \leq 2\sqrt{(1+\ln T)L_\infty\sum_{t=1}^{T}\langle\boldsymbol{L}_t, \boldsymbol{u}\rangle} + O(\ln T), \qquad \forall\boldsymbol{u}\in\triangle_d.$$