# OpenReview forum: "Adaptive Algorithm for Non-Stationary Online Convex-Concave Optimization"
_ICLR.cc/2025/Conference — Submitted to ICLR 2025_

### Official Review · Reviewer_rntL · 2024-10-18

**Soundness:** 2
**Presentation:** 3
**Contribution:** 2
**Rating:** 5
**Confidence:** 4

**Summary:**

This paper considers the problem of online convex-concave optimization, which can be seen as an adaption of online convex optimization (OCO) in the two-player game setup. For this problem, the authors first show that when both players run an Ader or Ader-like algorithm, the proposed dynamic duality gap can be bounded as the dynamic regret (which is a two-player version in the game setup). Consequently, the authors consider further enhancing the results with prediction-error-based guarantees, which are usually called optimistic online learning in the OCO setup. Finally, the authors investigate the case where only one prediction sequence (optimism sequence) could perform poorly and consider combining multiple optimism with learnable weights to obtain a more robust prediction sequence. By doing this, they can further enhance the results to the minimum of the prediction error of all possible optimism and the dynamic regret. Finally, empirical studies validate the effectiveness of the proposed method.

**Strengths:**

This paper considers a meaningful problem of achieving dynamic regret and optimistic bounds in the setup of two-player convex-concave games. The presentation is clear and easy to follow. Although I did not check the detailed proof, I believe that the proofs are correct thanks to the clear presentations by the authors.

**Weaknesses:**

My concern mainly lies in the technical contributions. Since I am kind of familiar with dynamic regret minimization and optimistic online learning, I found the technical contributions of this work somewhat limited.

1. The results in the adaptive module can be easily attained by using Ader for both players and the dynamic duality gap, in this case, can be decomposed into the summation of the dynamic regret of each player. The authors also acknowledge this.

2. For the integration module, although the equations on page 5 look complicated, they are mostly actually a two-player extension of the optimistic online mirror descent. For example, (3a) and (3d) are actually the optimistic step that uses the optimism ($H_t$ and $W_t$, $W_t$ is actually an optimism since it consists of the prediction $h_t$). And (3b), (3c) are the update step that uses the true loss information $f_t(\cdot)$. This is very similar to the one-player optimistic online mirror descent [1]. And Theorem 3, 4, 5 are not surprising to me since Theorem 3 is actually the optimistic bound, Theorem 4 is the regret bound attained by using Hedge (or clipped Hedge) as the meta-algorithm, and Theorem 5 combines the two theorems as mentioned earlier together.

3. The third multi-predictor aggregator is also not very surprising to me because it actually uses the technique of learning over multiple optimism. For example, the technique appears in the Section 4 "Learning The Predictable Processes" in [1]. As a result, this parts seems to be a direct extension from my point of view.

References:

1. Online Learning with Predictable Sequences, COLT 2013

**Questions:**

I have one question about the upper and lower bounds in the adaptive module. In my opinion, the dynamic regret measured by $P_T$ is meaningful since the variation of the $u_t$ and $v_t$ sequences reflect the difficulties of the learning problem. However, the quantity $C_T$ is less meaningful for me because $x_t^\prime$ actually depends on $y_t$ (also for $y_t^\prime$), showing that $C_T$ is in fact *algorithm-dependent*. As a result, one cannot justify how good the bound measured by $C_T$ is since $C_T$ relies on the algorithm.

Also, I agree that the lower bound for $P_T$ is correct, as explained by the authors after Proposition 2 and the lower bound indicated in the one-player setup. However, I am confused about the lower bound for $C_T$. As I said before, $C_T$ is algorithm-dependent. However, a lower bound should hold for all possible algorithms. Why could an algorithm-dependent quantity appear in a lower bound? Or maybe this lower bound only holds for this specific algorithm? In the latter case, the authors should revise the statement of Proposition 2 to state clearly that this lower bound only holds for Ader-like algorithms but not for all possible algorithms. I am looking forward to the authors' reply on this issue.

---

> ### Author Response · Authors · 2024-11-17
>
> Thank you for your valuable feedback and positive comments on our work's clarity and significance. We appreciate the opportunity to clarify our work.
>
> **Regarding the Technical Contributions**
>
> We understand your concern that some of the results might appear straightforward, as they build on existing concepts like dynamic regret minimization and optimistic online learning. However, we believe our contribution lies in the novel combination of optimism, implicit updates, and a modified meta-expert framework, which together provide a robust approach to the OCCO problem. Let me clarify our technical evolution:
>
> 1. To handle varying levels of non-stationarity, we employ a pair of ADER algorithms, where the expert layer algorithms run a set of OMDs. To achieve both minimax optimality and prediction error bounds, we need to replace OMD in the expert layer with some *optimistic and implicit* variant of OMD, which is currently unknown and represents a significant extension of previous work. Here, optimism introduces predictors, and implicit updates ensure prediction error bounds (see [2] for implicit update benefits over [1]).
>
> 2. For simplicity and to enhance computability, we pack this ADER pair into the Adaptive Module and analyze the optimistic implicit variant separately. We found that running Optimistic Implicit OMD independently on each player does not yield the desired prediction error bound; thus, we introduce a synchronized update using a time-varying Proximal Point Method for strategy generation:
> \begin{align}
> (\hat{x}\_t,\hat{y}\_t)&=\arg\min_{x\in X}\max_{y\in Y}h_t(x,y)+B_\phi (x, \tilde{x}\_t^\phi)/\eta_t-B_\psi (y, \tilde{y}\_t^\psi )/\gamma_t,
>  &&(1)\\\\
> \tilde{x}\_{t+1}&=\arg\min_{x\in X}\eta_t f_t(x,y_t)+B_\phi (x, \tilde{x}\_t^\phi), &&(2)\\\\
> \tilde{y}\_{t+1}&=\arg\max_{y\in Y}\gamma_t f_t(x_t,y)-B_\psi (y, \tilde{y}\_t^\psi),&&(3)
> \end{align}
> the synchronous update appears only in Eq.(1), while Eq.(2)&(3) independently update the auxiliary variables. This is an interesting insight we discovered during the proof.
>
> 3. The updates above, although effective in achieving prediction-error bounds, are not directly compatible with the meta-expert framework. Therefore, we modify Eq.(1)–(3) into a carefully constructed *joint update* that incorporates the meta-layer, as shown in Eq.(3a)-(3f), leading to the desired bound in Theorem 5. Additionally, we propose a solution framework for this joint update in Theorem 6. These are centrals to our contributions. Regarding the design details of the joint update, please refer to the response to Reviewer 1qNz [here](https://openreview.net/forum?id=WIerHtNyKr&noteId=reHg6XTjrO).
>
> 4. Finally, we extend multi-predictor support by using *clipped Hedge with adaptive learning rates*. We chose this algorithm because of the in-depth regret analysis presented in [3], which shows that a direct extension of [1] would not be sufficient.
>
> In contrast, the state-of-the-art algorithm in [4] can be viewed as a modified version of the ADER algorithm, where the expert layer algorithms must satisfy the DRVU property (e.g., Optimistic OMD), and the meta-layer uses Optimistic Hedge. The technical contribution lies in the construction of the meta-layer loss function. In terms of results, while the algorithm in [4] guarantees three discrete metrics, our approach focuses on D-DGap, covering all levels of nonstationarity, and while their algorithm doesn't guarantee tight bounds, ours ensures minimax optimality and tracks the most accurate predictor, further reducing the D-DGap.
>
> We hope the above clarifications address your concerns and help highlight the novel aspects of our contributions.
>
> [1] Online Learning with Predictable Sequences. COLT 2013\
> [2] Temporal variability in implicit online learning. NeurIPS 2020\
> [3] A closer look at temporal variability in dynamic online learning. arXiv:2102.07666\
> [4] No-Regret Learning in Time-Varying Zero-Sum Games. ICML 2022

---

> ### Author Response · Authors · 2024-11-17
> **Regarding the Role of $C_T$**
>
> **Regarding the Role of $C_T$**
>
> $C_T$ is not unique to the ADER algorithm but is a general quantity. Its dependence on the algorithm naturally arises in two-player settings. I would like to clarify this step-by-step:
>
> 1. In our definition of the dynamic duality gap (D-DGap), we introduce arbitrary comparator sequences $\\{u_t\\}$ and $\\{v_t\\}$, similar to the approach in dynamic regret where an arbitrary $u_t$ is used. This enables adaptability to a wide range of non-stationarity levels, as $\\{u_t\\}$ and $\\{v_t\\}$ can be freely chosen even after $T$ rounds. So the metric $P_T$ reflects a belief or an assumption about the environment's variability, rather than mirroring the precise level of non-stationarity observed.
>
> 2. After $T$ rounds, the environment has been fully observed, making only certain values of $P_T$ meaningful. Here, $C_T$ serves as an upper limit for $P_T$, based on the fact that $f_t(x_t, v_t) - f_t(u_t, y_t) \leq f_t(x_t, y'_t) - f_t(x'_t, y_t)$. Values of $P_T$ exceeding $C_T$ are irrelevant in this case, which is why $C_T$ appears in the upper bound. Likewise, $C_T$ also appears in the lower bound, as the lower bound depends on $P$, a preset upper limit for $P_T$. Setting $P$ above $C_T$ would make the bound excessively loose, so $C_T$ acts as the effective threshold.
>
> 3. While $C_T$ is algorithm-dependent in our two-player framework, this dependency reflects the mutual influence of strategies in the two-player setting, a feature fundamentally different from single-player setups. For example, in a single-player case, $C_T$ depends only on the environment, since $\ell_t(x_t) - \ell_t(u_t) \leq \ell_t(x_t) - \ell_t(x_t^*)$, where $x_t^* = \arg\min_x \ell_t(x)$. In two-player games, such dependencies are generally unavoidable, as player interactions influence strategy and hence the bound.
>
> While we recognize the typical preference in online learning for algorithm-independent bounds, we believe clarifying the role of $C_T$ is important. This is similar to how regret bounds involving $g_t=\nabla\ell_t(x_t)$ are algorithm-dependent, yet are commonly used in OCO [5]. Additionally, $C_T$ and $P_T$ appear together, with $P_T$ being algorithm-independent, which helps mitigate concerns about algorithm dependence to some extent.
>
> We have added a brief explanation of $C_T$ to our paper.
>
> [5] Orabona. A Modern Introduction to Online Learning. arXiv:1912.13213

---

> ### Author Response · Authors · 2024-11-23
>
> Dear Reviewer rntL,
>
> I hope this message finds you well. I wanted to kindly follow up regarding the responses I provided to your comments. As the rebuttal phase is nearing its deadline, I wanted to ensure that you had sufficient time to review my replies and ask if there is any additional clarification or information I can provide.
>
> Thank you again for your valuable feedback and for taking the time to review our work. I truly appreciate your efforts and look forward to hearing your thoughts.
>
> Best regards,\
> Authors

---

### Official Review · Reviewer_5tL9 · 2024-11-03

**Soundness:** 3
**Presentation:** 3
**Contribution:** 2
**Rating:** 5
**Confidence:** 2

**Summary:**

This paper addresses the problem of online convex concave optimization, where the objective is to minimize the dynamic duality gap. They propose a modular algorithm comprising of three components and guarantee a minimax optimal duality gap upper bound up to a logarithmic factor. They also provide some empirical results to further demonstrate
the performance of the proposed method.

**Strengths:**

S1 - The paper is generally well written and well presented.

S2 - The technical claims in the paper are generally well explained.

S3 - Experiments are a plus.

**Weaknesses:**

W1 - Comparisons to the existing literature are not enough. While the related works are comprehensive, their relation and comparison to your work is lacking.

W2 - The role of each module in the algorithm seems ambiguous at times.

W3 - The problem definition is poorly explained especially for a newly proposed performance metric.

**Questions:**

Q1 - Eq 1 is weird in that it is an equality but for every u_t and v_t, is it a function of them?

Q2 - While i understand that the duelity gap is an important metric, its role as a performance metric seems dubious.

Q3 - Where in the algorithm does your main novelty reside? It mainly seems like an amalgamation of existing techniques.

---

> ### Author Response · Authors · 2024-11-18
>
> Thank you for your valuable feedback and positive comments on our work's clarity. We appreciate the opportunity to clarify our work.
>
> **Response to W1: Comparison to Existing Literature**
>
> While the literature on OCO and convex-concave optimization is extensive, studies specifically addressing OCCO remain limited. Among them, the most related work is Zhang et al. (2022), which we discuss in Lines 61-64 to highlight key limitations and motivate our work.
>
> Our work surpasses Zhang et al. in two major ways:
>
> 1. **Unified Metric:** Zhang et al. evaluate three separate metrics (static individual regret, duality gap, and dynamic Nash equilibrium regret), while we propose the dynamic duality gap (D-DGap) as a unified metric, capable of capturing all levels of nonstationarity.
>
> 2. **Tighter Bounds:** While Zhang et al. lacks tightness guarantees, our algorithm achieves minimax optimality against arbitrary comparator sequences, effectively tracking the best predictor and further reducing the D-DGap.
>
> We have revised the **Related Work** section to provide a clearer comparison with Zhang et al., emphasizing these advancements.
>
> **Response to W2: The Roles of Three Modules**
>
> The roles of the three modules are explained in detail in our paper:
>
> 1. *Overview of Roles:* In Lines 080-092, we describe the functionality of each module and briefly outline their design principles as part of introducing our contributions.
>
> 2. *Detailed Explanation:* Section 4.5 is dedicated to explaining how each module functions and how they work together to form the overall algorithm.
>
> We believe these sections clarify the distinct roles and collaborative interactions of the modules.
>
> **Response to W3, Q1, Q2: D-DGap**
>
> Let me explain the problem definition and clarify the role of D-DGap as a metric in OCCO to address your concerns.
>
> *Relationship Between OCCO and OCO*
>
> OCCO extends OCO by introducing two interacting players engaged in time-varying convex-concave games. While OCO focuses on minimizing convex losses for a single player, OCCO models the dynamic interplay between two players -- one minimizing and the other maximizing a shared convex-concave payoff function. OCCO can be reduced to OCO if we constrain the action set of one player to a singleton.
>
> *D-Reg in OCO*
>
> Dynamic Regret (D-Reg) evaluates a single player's performance by comparing cumulative losses to those of an arbitrary comparator sequence. Formally:
> \\[
> \text{D-Reg}\_T=\sum_{t=1}^T\ell_t(x_t)-\sum_{t=1}^T\ell_t(u_t),\quad\forall u_t\in X.
> \\]
> Here, $\\{u_t\\}$ reflects a belief or an assumption about the environment's variability. For example:
> - If $u_t = x_t^*=\arg\min_x \ell_t(x)$, D-Reg benchmarks against dynamically optimal strategies.
> - If $u_t$ is fixed, it assumes a stationary environment.
>
> The flexibility of $u_t$ allows D-Reg to accommodate diverse scenarios, and restrictions on its path length $P_T$ are often imposed to avoid vacuous bounds in hostile environments. Although D-Reg is formally a function of $u_t$, it is sometimes presented without explicitly stating $u_t$, as seen in Line 035, for simplicity.
>
> *D-DGap in OCCO*
>
> Dynamic Duality Gap (D-DGap) extends the classical duality gap to benchmark performance against arbitrary comparator sequences $(u_t,v_t)$, a natural generalization of the D-Reg in OCO. Formally:
> \\[
> \text{D-DGap}\_T=\sum\_{t=1}^T f_t(x_t,v_t)-\sum\_{t=1}^T f_t(u_t,y_t),\quad\forall(u_t, v_t)\in X\times Y.
> \\]
> While the classical duality gap chooses $x'\_t=\arg\min\_{x\in X}f_t(x,y_t)$ and $y'\_t=\arg\max\_{y\in Y}f_t(x_t,y)$ as worst-case comparators at time $t$, the D-DGap uses $(u_t,v_t)$ to account for a broad spectrum of nonstationary levels. This design mirrors the role of $u_t$ in D-Reg. D-DGap is a function of $(u_t, v_t)$, but the comparators are omitted for brevity in the definition (Eq.(1), Line 057).
>
> *Why D-DGap is Meaningful*
>
> Both sublinear D-Reg and sublinear D-DGap reflect approximate coarse correlated equilibria. Against dynamically optimal comparators, sublinear D-Reg ensures approximate dynamic optimality, while sublinear D-DGap indicates near Nash equilibrium in most iterations. When compared to stationary comparators, both align with traditional game-theoretic settings. D-DGap naturally extends D-Reg by evaluating two-player interactions, offering a unified framework for performance assessment in OCCO.
>
> **Response to Q3: Novelty**
>
> The main novelty of our algorithm lies in its modular design, particularly the **joint update mechanism**, which has not been explored in the existing literature.
>
> For details on the **Technical Contributions**, please refer to our response to Reviewer rntL [here](https://openreview.net/forum?id=WIerHtNyKr&noteId=2YGCeRBRKF).
>
> For specifics on the **joint update mechanism**, please see our response to Reviewer 1qNz [here](https://openreview.net/forum?id=WIerHtNyKr&noteId=reHg6XTjrO).

---

> ### Author Response · Authors · 2024-11-23
>
> Dear Reviewer 5tL9,
>
> I hope you are doing well. I’m writing to kindly check in regarding my responses to your comments. With the rebuttal period approaching its deadline, I wanted to ensure that everything is clear and address any remaining concerns or questions you might have.
>
> Thank you once again for your valuable feedback and for your time in reviewing our work. I look forward to your thoughts.
>
> Best regards,\
> Authors

---

> > ### Comment · Reviewer_5tL9 · 2024-11-27
> >
> > Thank you for the clarifications. I have also read the other reviews and rebuttals. I do not have any further questions.

---

### Official Review · Reviewer_1qNz · 2024-11-04

**Soundness:** 3
**Presentation:** 3
**Contribution:** 3
**Rating:** 6
**Confidence:** 3

**Summary:**

The paper considers a "dynamic duality gap" objective for online convex-concave optimization (carrying the OCO dynamic regret definition over to the OCCO setting in a natural way). It derives an algorithm which both achieves the near-minimax dynamic regret bound for this problem, and achieves near-constant regret when the setting is stationary. More generally, the algorithm is capable of taking in multiple predictions of the objective function and achieving performance close to the best one (while maintaining its minimax dynamic regret bound). The efficacy of the algorithm is verified on several synthetic experiments, and is compared --- with favorable results --- against the performance existing time-varying game-solving method of Zhang, Zhao, Luo, and Zhou (NeurIPS 2022).

**Strengths:**

Overall, I believe that this is a good contribution to the literature on solving convex-concave time-changing games/saddle point problems.

I tend to agree with the authors' stated assessment that given the previous state-of-the-art time-varying game solving method of Zhang, Zhao, Luo, and Zhou, it is a-priori not trivial to e.g. incorporate the recent advances in optimistic adaptive/dynamic algorithms to improve Zhang et al's method's performance in stationary/not fully adversarial environments while obtaining the minimax dynamic regret guarantees. This paper addresses this challenge by developing a natural algorithmic scheme with good dynamic regret performance and the ability to obtain optimal performance in stationary settings.

In terms of the overall assessment on the technical side, in my opinion the main technical contribution of the paper can be found in its "integration module", which specifies how to reduce the aggregation of the dynamic regret predictions and other available predictions to a fixed point problem. The "adaptive module" and "the multi-predictor aggregator" are useful but relatively generic components, and the main insight in relation to these two modules is that porting them from the OCO setting into the OCCO setting ends up giving meaningful adaptive/dynamic guarantees in this setting as well. Overall, to sum up the strong sides of this paper:

+ By appropriately formulating the dynamic duality gap definition, the paper demonstrates how to appropriately port OCO dynamic regret/meta-aggregation insights into the more involved OCCO setting, without losing the tight minimax rate of dynamic regret algorithms like ADER from the OCO setting.

+ The authors put in the effort to state the framework in a modular way: they separate out the parts where an OCO dynamic regret algorithm like ADER, an expert-aggregation algorithm, and a fixed-point algorithm are applied. This may help with future refinements of, and practical improvements to, the analysis.

**Weaknesses:**

The paper appears to provide sound statements and proofs, and is overall solidly written. That being said, I am a bit tentative/unconvinced regarding the following several points, which I would appreciate if the authors could clarify:

- In regards to the regret bound terms that involve \rho (i.e., the terms that promise performance not worse than that of the predictors h), the way \rho is currently defined is as the max-norm deviation between h and f over all arguments (x, y). While this is of course already a nontrivial guarantee, but it seems a bit crude, and I believe the result would be strengthened, and made more OCCO-compatible, if a saddle-point specific metric of distance between f and h were considered (e.g., something along the lines of dist(h, f) = max{max_x |h(x, y*(x)) - f(x, y*(x))|, max_y |h(x*(y), y) - f(x*(y), y)|}, where x*(y) is the best-response of the first player to y and where y*(x) is the second player's best-response to x). A quick glance at eq (10) in the appendix suggests that this would not be an easy change to make in the analysis (and in fact even tightly replacing the metric \rho by an L1 or L2 style metric looks like it wouldn't be trivial either?).

- In terms of the modular framework itself, I am still not fully convinced about the arguments that were presented in-text about why the meta-aggregation mechanism (i.e. the "integration module") had to be designed in this bespoke way where the meta-layer variables v, w have to interact (in the interdependent update equations (3a) and (3d)) with the predictor(s) h that are being aggregated, and --- moreover --- these predictors h themselves have to interact with the adaptive module predictions \bar{x} and \bar{y}. Why is it impossible to break both of these interdependencies to make things completely modular --- e.g., along the lines of turning equation (3a) into the following form free of both the w, omega, and of the \bar{x}_t, \bar{y}_t variables: (\hat{x}_t, \hat{y}_t) = arg min_x max_y h_t(x, y)  + <regularization terms with B_\phi and B_\psi>?

- The experimental section overall was designed reasonably well, but a few aspects merit addressing/clarifying. (a) It says: "our algorithm consistently outperforms Zhang et al", which appears to be an overstatement: In cases I and III, the performance of both methods on the D-DGap plot is very similar, and only in case II do we see a significant difference (for all three levels of nonstationarity). Given the stronger guarantees of the new algorithm, I don't doubt that it can outperform Zhang et al's in many settings, but to make that point well-substantiated, more experiments need to be shown. (b) Also, for the sake of transparency, I am wondering how much more compute time does the new algorithm use compared to Zhang et al's? (I do not consider it strictly necessary to report this in the paper, though if the runtime is somehow comparable (which I don't a-priori expect due to the bulky procedure in each round) then mentioning this in the manuscript would of course be favorable to the current paper.) (c) Is there any meaning behind the 1/ ln(t+1) scaling in the level-ii comparator?

**Questions:**

Please see the several questions integrated in the weaknesses section above.

---

> ### Author Response · Authors · 2024-11-17
>
> Thank you for your thoughtful evaluation of our contributions and your insightful feedback. Your comments have been invaluable in helping us identify areas requiring further clarification and additional experiments. We have carefully addressed each of your concerns as outlined below.
>
> **Regarding the Terms Involving $\rho$ and the Suggested Metric**
>
> We define $\rho$ as the max-norm deviation between the predictor $h_t$ and the payoff function $f_t$ over all arguments $(x, y)$. This choice allows us to derive clean and analytically tractable D-DGap bounds. While $\rho(f_t, h^k_t)$ may be considered a relatively crude metric, it can be refined as
> \\[
> L_t^k = \max_{x\in\\{\hat{x}\_t,\bar{x}\_t,\tilde{x}\_{t+1}\\},y\in\\{\hat{y}\_t,\bar{y}\_t,\tilde{y}\_{t+1}\\}} \lvert f_t(x,y) - h_t^k(x,y)\rvert.
> \\]
> This refinement offers computational convenience but introduces intermediate variables that make it unsuitable for inclusion in the upper bound.
>
> We appreciate your suggestion to replace $\rho$ with a refined metric based on the saddle-point structure of the problem. Although such a metric could potentially yield tighter bounds, adopting it would require significant changes to the framework, including revisions to the regret decomposition and update rules, complicating the proof structure. While this alternative is theoretically appealing, we believe it falls outside the scope of this work, which is focused on a straightforward approach for OCCO. We will consider more refined metrics in future work.
>
> **Regarding the Coupling of the Meta-Layer and the First Expert**
>
> The coupling between the meta-layer and the first expert is intentionally designed to address the specific requirements of our framework. Let's first explore what happens if these components were decoupled.
>
> If the first expert were not influenced by the meta-layer's weights, its updates could be written as:
> \begin{align}
> (\hat{x}\_t,\hat{y}\_t)&=\arg\min_{x\in X}\max_{y\in Y}h_t(x,y)+B_\phi (x, \tilde{x}\_t^\phi)/\eta_t-B_\psi (y, \tilde{y}\_t^\psi )/\gamma_t,
>  &&(1)\\\\
> \tilde{x}\_{t+1}&=\arg\min_{x\in X}\eta_t f_t(x,y_t)+B_\phi (x, \tilde{x}\_t^\phi), &&(2)\\\\
> \tilde{y}\_{t+1}&=\arg\max_{y\in Y}\gamma_t f_t(x_t,y)-B_\psi (y, \tilde{y}\_t^\psi).&&(3)
> \end{align}
> The instantaneous D-DGap would then decompose as follows:
> \begin{align}
> f_t(x_t,v_t)-f_t(u_t, y_t)
> = &f_t(x_t, v_t)-f_t(x_t,\tilde{y}\_{t+1})+f_t(\tilde{x}\_{t+1}, y_t)-f_t(u_t, y_t)&&(4)\\\\
> &+f_t(x_t,\tilde{y}\_{t+1})-f_t(\hat{x}\_t,\tilde{y}\_{t+1})+f_t(\tilde{x}\_{t+1},\hat{y}\_t)-f_t(\tilde{x}\_{t+1}, y_t)&&(5)\\\\
> &+f_t(\hat{x}\_t,\tilde{y}\_{t+1})-h_t(\hat{x}\_t,\tilde{y}\_{t+1})+h_t(\tilde{x}\_{t+1},\hat{y}\_t)-f_t(\tilde{x}\_{t+1},\hat{y}\_t)&&(6)\\\\
> &+h_t(\hat{x}\_t,\tilde{y}\_{t+1})-h_t(\hat{x}\_t,\hat{y}\_t)+h_t(\hat{x}\_t,\hat{y}\_t)-h_t(\tilde{x}\_{t+1},\hat{y}\_t).&&(7)
> \end{align}
> Here:
> - The upper bound for Eq.(4) can be obtained using the first-order optimality conditions of Eq.(2)&(3).
> - The upper bound for Eq.(7) follows from the first-order optimality conditions of Eq.(1).
> - Eq.(6) represents the prediction error.
> - Eq.(5) implies the meta-regret.
>
> However, this decomposition is only valid for the first expert. Specifically, the meta-layer described in Eq.(5) is incompatible with the second expert (the Adaptive Module). Furthermore, Eq.(5) suggests that the meta-layer loss depends on $f_t$, $\tilde{x}\_{t+1}$, and $\tilde{y}\_{t+1}$. To achieve a meaningful prediction error bound, the meta-layer must not only rely on the predictor $h_t$, but also predict $\tilde{x}\_{t+1}$ and $\tilde{y}\_{t+1}$, ultimately expressing all prediction errors in terms of the gap between $f_t$ and $h_t$. This presents significant challenges in algorithmic design.
>
> Given the above, let me recap how the joint update was designed:
>
> 1. For the meta-layer of the integration module, it needs to satisfy two requirements:
>     1. It must be compatible with both experts simultaneously.
>     2. It must achieve a prediction error-type upper bound while maintaining minimax optimality.
>
> 2. We prove that the update rule for the meta-layer, defined by Eq.(3d)-(3f), satisfies the second requirement. Specifically, Eq.(3d) is critical in ensuring the prediction error-type upper bound. Up to this point, we have not yet addressed the interaction between the meta-layer and the experts.
>
> 3. The meta-layer, as defined by Eq.(3d)-(3f), is compatible with the second expert (refer to Line 291). We now need to ensure that the first expert is also compatible with the meta-layer, and that its update rule shares similar characteristics with those in Eq.(1)-(3). This is important because its upper bound contains the prediction error term from Eq.(6). Taking both factors into account, the decomposition for the first expert should follow the structure in Lines 843-847 (previously 789-793), and its update rule must adhere to Eq.(3a)-(3c). This accounts for the rationale behind designing the joint update mechanism.

---

> ### Author Response · Authors · 2024-11-17
>
> Due to the addition of new experiments, some figure numbers have changed. We use the updated numbering in this response, with the previous numbering indicated in parentheses.
>
> **a. Regarding the Experimental Results**
>
> Case I represents a gradually stabilizing environment. When a stable comparator sequence is used, both our algorithm and Zhang et al.'s algorithm achieve near-$\widetilde{O}(1)$ D-DGap bounds, as shown in Fig.2a, the performances are nearly identical in this setting. In contrast, when a highly non-stationary comparator sequence is used, our algorithm maintains a near-$\widetilde{O}(1)$ bound, while Zhang et al.'s algorithm degrades to a near-$\widetilde{O}(\sqrt{T})$ bound. This is shown in Fig.2c (previously 2g), where our algorithm outperforms theirs. The advantage may seem less apparent due to the log-log scale and the use of average D-DGap per round on the vertical axis. However, the advantage grows as the game progresses.
>
> Case IV (previously III) models an adversarial environment. As shown in Fig.2j, 2k, and 2l (previously 2c, 2f, and 2i), the performances of both algorithms are very similar. This is because our algorithm guarantees minimax optimality, while Zhang et al.'s algorithm, despite lacking tight bounds, benefits from the meta-expert framework, leading to performance close to minimax optimality.
>
> We acknowledge that the original phrasing in the manuscript may have caused some confusion. To address this, we have revised the manuscript with clearer language.
>
> Additionally, we included a new experimental scenario to further validate the advantages of our algorithm. In Case III, the saddle point cyclically shifts among seven distinct branches, with its position within each branch determined randomly. Compared to Case II, this setup introduces additional randomness, making precise predictions infeasible. Fig.2g, 2h, and 2i illustrate that, even under these challenging conditions, our algorithm exhibits a clear performance advantage.
>
> **b. Regarding Computational Time**
>
> In our experiments, the average computation time per round for our algorithm and Zhang et al.'s algorithm was 2.33ms and 0.071ms, respectively, under identical experimental conditions. The increased runtime of our algorithm stems from the additional complexity introduced by its design.
>
> Zhang et al.'s algorithm can be viewed as a modified version of the ADER algorithm, where the expert layer satisfies the DRVU property (e.g., Optimistic OMD) and the meta-layer utilizes Optimistic Hedge. In theory, its computational cost is marginally higher than running the Adaptive Module alone. By contrast, our algorithm dedicates the majority of its computational resources to joint updates, which deliver stronger theoretical guarantees. These guarantees include adaptability to diverse non-stationary environments and compatibility with multiple predictors.
>
> As we noted in the Conclusion section: *"A promising direction for future research is to explore a balance between relaxing strict guarantees on the D-DGap bound and reducing computational costs, enabling a better trade-off for time-sensitive applications."*
>
> **c. Regarding the $1/\ln(t+1)$ Scaling**
>
> Level i corresponds to a stable comparator sequence with a path length of $P_T = 0$. Level iii represents a highly non-stationary comparator sequence, where the path length $P_T$ grows linearly over time. Level ii lies between these extremes, with the $1/\ln(t+1)$ scaling ensuring sublinear growth of $P_T$. Notably, $1/\ln(t+1)$ is not the only possible scaling factor; alternative choices may also be valid.

---

> ### Author Response · Authors · 2024-11-23
>
> Dear Reviewer 1qNz,
>
> I hope you are doing well. As the rebuttal deadline approaches, I wanted to check in regarding the responses I provided to your comments. I would be glad to elaborate further if you have any additional questions or need further clarifications.
>
> Thank you for your valuable time and feedback throughout this process. I sincerely appreciate your effort in reviewing our submission.
>
> Best regards,\
> Authors

---

> > ### Comment · Reviewer_1qNz · 2024-11-25
> >
> > Thank you for the detailed rebuttal. The experimental section, with the requested clarifications performed and with an extra experimental case of outperforming Zhang et al (2022) added, is now in a better shape. In the other sections, the contributions and some further technical details (e.g. about C_T) have been clarified.
> >
> > I have also read the other reviews and your rebuttals to those, which are similarly quite detailed and constructive. I overall keep my "weak accept" assessment of the manuscript. If I were to summarize the overall comments from my and the other reviews at the current stage:
> >
> > (1) The design and analysis of the proposed algorithm, in conjunction with the introduced D-DGap metric, is quite modular and well-explained, which is a significant plus from my perspective. It allows to e.g. easily swap in components controlling adaptability to dynamic environments, and also components that add in "side knowledge" from other predictors. Multiple reviews, on the other hand, have emphasized the fact that the technical contribution is not groundbreaking. Indeed, the analysis mostly consists of careful modifications to existing techniques familiar in the online learning literature, with the exception of main technical contribution, which is the proposal to inject some coupling into the meta-expert layer (which is e.g. detailed by the authors in the rebuttal to my review). Also, my criticism of the "insufficiently groundbreaking" flavor points out that a more interesting/insightful metric rho of distance to other predictors could be considered. But, on the balance, I believe that many refinements of this type can be justifiably left for future work, and that the modularity of the manuscript's algorithm and writing is a strength that compensates for the moderate amount of technical novelty.
> >
> > (2) The experimental results were not fully convincing to me and some other reviewers, as the shown margin relative to Zhang et al was quite slim in most shown cases. The update after the rebuttals has somewhat improved the situation, though the picture remains nuanced, which however is not unnatural per se; indeed, the OCCO environment is itself complex and nuanced, plus Zhang et al's algorithm, owing to its principled design, can definitely show better performance than its worst-case guarantees on a case-by-case basis.
> >
> > (3) The writing of the paper is quite nice structure-wise, and that has been pointed out by my and some other reviewers. However, some specifics that are particular to the setting of OCCO, namely the form of the path-length-type quantities appearing in the bounds, perhaps needed to be explained in a more reader-friendly way, hence why the questions on that appeared in the other reviews (and have been reflected in the rebuttal responses and in the updated manuscript by the authors). Ditto for my concerns about the explanation for why the special "entangled" structure of the meta-step is needed (and the rebuttal to my review addressed that). On the balance, however, my opinion of the presentation quality is positive.

---

> > > ### Author Response · Authors · 2024-11-28
> > >
> > > Thank you for your comment. We will further refine our manuscript and include additional comparative experiments to incorporate standard OCO algorithms as baselines.

---

### Official Review · Reviewer_uyS2 · 2024-11-04

**Soundness:** 2
**Presentation:** 2
**Contribution:** 2
**Rating:** 5
**Confidence:** 3

**Summary:**

Briefly, this paper studies the problem of Online Convex-Concave Optimization (OCCO) and contributes improved dynamic duality gap (D-DGap) performance bound, which is the counterpart of dynamic regret in the original Online Convex Optimization (OCO) problem. Their setting considers stationary, non-stationary and predictable environments. Their approach includes an Adaptive Module for non-stationarity, a Multi-Predictor Aggregator for predictability, and an Integration Module for combination.

**Strengths:**

**Originality:**
- The work is somewhat a novel combination of well-known techniques such as predictive updates, expert combination and dynamic learning.

**Quality:**
- The submission is generally technically sound.
- The claims are supported by theoretical analysis.
- The methods used are appropriate.
- This is a complete piece of work, with potential for further improvements.

**Clarity:**
- The submission is generally clearly written.
- It is mostly well organized.

**Significance:**
- The others (researchers or practitioners) likely to use the ideas or build on them, since it has improved guarantees.
- It provides new conclusions through a somewhat unique theoretical approach of joint optimization.

**Weaknesses:**

**Originality:**
- The tasks and methods are not new, and the exact novelties are unclear.
- It is not clear how this work differs from previous contributions as the challenges in achieving the results are mostly missing.
- The related work may not be adequately cited as the experiments lack substantial comparisons.

**Quality:**
- The authors should be more careful about evaluating the strengths and weaknesses of their work, especially in the experiments.

**Clarity:**
- It does not adequately inform the reader to reproduce the experimental results.

**Significance:**
*(Key Weakness)*
- The submission does not seem to address a difficult task in a better way than previous work. The hardness of the task and improvement over the literature could be much better explained.
- The results seem to be lackluster in importance, as the difference from the existing literature seems incremental.
- It fails to demonstrate how it advances the state of the art in the experimental analysis.

**Questions:**

**Questions:**
- Is your main novelty the joint optimization in (3)?
- Line 59: what exactly do you mean by non-stationarity?
- Line 67: what do you mean by approximating? This and the following parts need clearer wording.
- Line 86: shape?
- Line 288: where did $O(T)$ term disappear, and also where did the coefficient $16$ come from?

**Major Suggestions:**
- Line 199: the value of $C_T$ in Proposition 2 is ambiguous, given the nature of $x_t$, $y_t$ in minimax analysis. The subsequent intuition is also consequently lacking. Please correct these.
- Line 505: in Figure 2, the difference with the existing literature seems negligible besides Case II, where your algorithm has advantages after utilizing appropriate predictors. Please explain the shortcomings of the existing literature.

**Minor Suggestion:**
- Line 98: if $C_T$ is a bound on $P_T$, why is it incorporated in this bound?
- Line 311: Clarify what the $\mathcal{F}_t$ function represents.

---

> ### Author Response · Authors · 2024-11-19
>
> Thank you for your feedback and for taking the time to review our submission. We appreciate the opportunity to address your concerns and clarify potential misunderstandings regarding our work. Below, we respond to your comments point by point.
>
> >*Originality: The tasks and methods are not new, and the exact **novelties** are unclear.*
>
> We respectfully disagree with your assessment that our work lacks novelty. Our key contributions, which distinguish this work from prior studies, are as follows:
> - **D-DGap:** Unlike previous works, we introduce the *dynamic duality gap (D-DGap)* as a generalized performance metric that bridges the gap between static individual regret and duality gap. This is a fundamental departure from the fragmented metrics used in prior literature, such as Zhang et al.(2022).
> - **Modular Framework:** Our algorithm adopts a modular design with a *joint update mechanism* that has not been explored before. This mechanism ensures the best of both worlds of minimax optimality and prediction accuracy.
> - **Tightness of Bounds:** Our algorithm achieves minimax optimality against arbitrary comparator sequences, effectively tracks the best predictor, and further reduces the D-DGap, an achievement not realized by previous works in OCCO.
>
> To further emphasize these points, we have revised the **Related Work** section to highlight these advancements.
>
> >*Originality: It is not clear how this work differs from previous contributions as the **challenges** in achieving the results are mostly missing.*
>
> OCCO introduces additional complexities compared to OCO due to the dynamic interplay between two players. Each player's decisions are influenced not only by the evolving environment but also by the strategy of their counterpart.
>
> The previous work by Zhang et al.(2022) can be viewed as a modified version of the ADER algorithm, where the expert layer algorithms must satisfy the DRVU property (e.g., Optimistic OMD), and the meta-layer uses Optimistic Hedge. The technical contribution lies in the construction of the meta-layer loss function.
>
> Unlike the approach of Zhang et al.(2022), which directly modifies the ADER algorithm, our method adopts a modular design, packaging the ADER algorithm into an Adaptive Module. Our primary challenge lies in the design of the *Integration Module* (Lines 210-212). Specifically, this module must:
> - Adapt to arbitrary comparator sequences.
> - Achieve a prediction error-based D-DGap bound.
>
> For a detailed explanation of the technical contributions and how they differ from prior work, we kindly refer you to our response to Reviewer rntL [here](https://openreview.net/forum?id=WIerHtNyKr&noteId=2YGCeRBRKF). For specifics on the **joint update mechanism**, please see our response to Reviewer 1qNz [here](https://openreview.net/forum?id=WIerHtNyKr&noteId=reHg6XTjrO).
>
> >*Originality: The related work may not be adequately cited as the **experiments** lack substantial comparisons.*\
> >*Quality: The authors should be more careful about evaluating the strengths and weaknesses of their work, especially in the experiments.*
>
> - **Citations:** We have cited all relevant prior works to the best of our knowledge (Lines 104–124). If there are specific works we may have overlooked, we would greatly appreciate further guidance. We recognize the importance of contextualizing our contributions within the existing literature and have added comparisons with Zhang et al.(2022) in the **Related Work** section to better highlight our advancements.
> - **Comparative Experiments:** We acknowledge that the original phrasing in the manuscript may have led to some confusion regarding the comprehensiveness of our comparisons. To address this, we have revised the manuscript (Lines 478–485) for clarity. The experimental section now provides detailed comparisons with the baseline algorithm proposed by Zhang et al.(2022), highlighting both theoretical and practical improvements. To further substantiate the advantages of our algorithm, we have included **a new experimental scenario (Case III)** to complement the existing evaluations.
>
> >*Clarity: It does not adequately inform the reader to reproduce the experimental results.*
>
> We respectfully disagree with your assessment that our results are not reproducible. To ensure reproducibility, we have:
> - **Released Source Code:** Please refer to the Supplementary Material.
> - **Implementation Details:** Section 5 (lines 428-476) of the manuscript outlines the settings used in our experiments.
>
> >***Significance:** (Key Weakness)*
> >- *The submission does not seem to address a difficult task in a better way than previous work. The hardness of the task and improvement over the literature could be much better explained.*
> >- *The results seem to be lackluster in importance, as the difference from the existing literature seems incremental.*
> >- *It fails to demonstrate how it advances the state of the art in the experimental analysis.*
>
> Please refer to the above responses.

---

> ### Author Response · Authors · 2024-11-19
>
> >*Questions: Is your main novelty the joint optimization in (3)?*
>
> Yes, the joint update mechanism in Eq.3 is the main innovation, but our contributions extend beyond that. Please refer to the above responses.
>
> ---
>
> >*Questions: Line 59: what exactly do you mean by non-stationarity?*
>
> Non-stationary OCCO refers to a setting where the environment evolves dynamically over time and cannot be treated as static. This concept is consistent with its definition in OCO.
>
> ---
>
> >*Questions: Line 67: what do you mean by approximating? This and the following parts need clearer wording.*
>
> The minimax optimal D-Reg bound is $O(\sqrt{(1 + P_T)T})$. However, the state-of-the-art algorithm achieves an $\widetilde{O}(\sqrt{(1 + P_T)T})$ D-Reg upper bound, which differs from the minimax optimal bound by a logarithmic factor hidden in the $\widetilde{O}$ notation. Ignoring logarithmic factors, we can roughly state that an $\widetilde{O}(\sqrt{(1 + P_T)T})$ bound achieves minimax optimality. More precisely, this bound approximates the minimax optimality.
>
> ---
>
> >*Questions: Line 86: shape?*
>
> shape -> sharp. Thank you for pointing it out.
>
> ---
>
> >*Questions: Line 288: where did $O(T)$ term disappear, and also where did the coefficient 16 come from?*
>
> The bound in Theorem 3 is
> \\[O\left(\sum_{t=1}^{T}\rho(f_t,h_t)\right),\\]
> and the bound in Theorem 4 is
> \\[O\left(\min\left\\{\sum_{t=1}^{T}\rho(f_t,h_t),\\sqrt{(1+\ln T)T}\right\\}\right)\leq O\left(\sum_{t=1}^{T}\rho(f_t,h_t)\right).\\]
> Combining these two results gives
> \\[\text{D-DGap}\_T\leq O\left(\sum_{t=1}^{T}\rho(f_t,h_t)\right).\\]
> In the original manuscript, we wrote
> \\[\text{D-DGap}\_T\leq O(1)+16\sum_{t=1}^{T}\rho(f_t,h_t).\\]
> This conclusion is verified in the proof details in the appendix, where the constant $16$ is the sum of contributions: $4$ (Line 906, previously 852), $4$ (Line 910, previously 856), and $8$ (Line 1023, previously 969).
>
> Considering the context, we have revised our manuscript. Thank you for your reminder.
>
> ---
>
> >*Major Suggestions: Line 199: the value of $C_T$ in Proposition 2 is ambiguous, given the nature of $x_t$, $y_t$ in minimax analysis. The subsequent intuition is also consequently lacking. Please correct these.*
>
> For details on **Regarding the Role of $C_T$**, please refer to our response to Reviewer rntL [here](https://openreview.net/forum?id=WIerHtNyKr&noteId=WsTaRpRPgU).
>
> We have updated our manuscript to include an intuitive explanation (Lines 205-207).
>
> ---
>
> >*Major Suggestions: Line 505: in Figure 2, the difference with the existing literature seems negligible besides Case II, where your algorithm has advantages after utilizing appropriate predictors. Please explain the shortcomings of the existing literature.*
>
> While Zhang et al.(2022) performs comparably in some experimental scenarios, their approach lacks the theoretical rigor to guarantee tight bounds and cannot integrate predictors effectively. In contrast, our algorithm achieves tighter bounds and demonstrates superior performance in highly dynamic settings (e.g., Case II and III). We have revised Lines 478–485 to clarify this distinction.
>
> ---
>
> >*Minor Suggestion: Line 98: if $C_T$ is a bound on $P_T$, why is it incorporated in this bound?*
>
> The concepts of $C_T$ and $P_T$ serve distinct purposes:
> - $P_T$ represents a belief or assumption about the level of non-stationarity of the environment. To maximize adaptability to dynamic changes, $P_T$ can follow any pattern, ranging from $0$ to linear growth.
> - $C_T$, in contrast, reflects an evaluation of the actual non-stationarity observed, based on the interaction between the two players and the environment. Importantly, $C_T$ acts as an upper bound for $P_T$. Retrospective analysis of the game history confirms that only those $P_T$ values within the limit of $C_T$ are consistent with the true dynamics of the environment.
>
> ---
>
> >*Minor Suggestion: Line 311: Clarify what the $\mathcal{F}\_t$ function represents.*
>
> $\mathcal{F}\_t$ is the objective function of the optimization problem induced by the joint update mechanism. Minimizing $\mathcal{F}\_t$ is equivalent to executing the joint update. Its mapping chain is illustrated as follows:
> \\[
> \mathcal{F}\_t: (w, \omega) \to (\boldsymbol{w}, \boldsymbol{\omega}) \xrightarrow{\mathcal{H}\_t} (\hat{x}, \hat{y}) \to \sigma,
> \\]
> where $\sigma = \max_{\boldsymbol{v}} W_t(\boldsymbol{w}, \boldsymbol{v}; \hat{x}, \hat{y}) - \min_{\boldsymbol{u}} W_t(\boldsymbol{u}, \boldsymbol{\omega}; \hat{x}, \hat{y})$.
>
> ---
>
> We hope this response addresses your concerns and clarifies the contributions and significance of our work. We sincerely appreciate your constructive feedback and look forward to any further suggestions you may have.

---

> ### Author Response · Authors · 2024-11-23
>
> Dear Reviewer uyS2,
>
> I hope this message finds you well. I wanted to gently follow up on my responses to your thoughtful comments. As the discussion phase is nearing its conclusion, I would be happy to provide further clarifications if needed. Please let me know if there are any aspects you'd like me to address in more detail.
>
> Thank you very much for your time and consideration. I greatly appreciate your effort in reviewing our submission.
>
> Best regards,\
> Authors

---

> ### Comment · Reviewer_uyS2 · 2024-11-27
>
> Thank you for the rebuttal.
>
> Your experiments are lacking with regards to comparisons from the literature, with Zhang et al. (2022b) being singled out. In the existing experiments, your set of comparisons could have included staple online optimizers, other OCCO algorithms and pairs of OCO algorithms from your relevant prior works. Since you observe $f_t$, then $x_t$ and $y_t$ could have been generated via $f_t(\cdot,v_t')$ and $-f_t(u_t',\cdot)$ for various sequences of $v_t'$ and $u_t'$, respectively, using D-Reg optimizers.
>
> Additionally, please note that Definitions 1 and 2 do not exist but there are Assumptions 1 and 2.

---

> > ### Author Response · Authors · 2024-11-28
> >
> > Thank you for your reply. We will include comparative experiments in future versions.
> >
> > Thank you for pointing out the issue, we have corrected the citation error caused by LaTeX's cleveref package.

---

### Meta-Review · Area_Chair_P2xR · 2024-12-11

**Metareview:**

This paper addresses Online Convex-Concave Optimization (OCCO), proposing a modular algorithm that minimizes the dynamic duality gap with minimax dynamic regret guarantees. The method includes an adaptive module, multi-predictor aggregator, and integration module, and is validated through synthetic experiments.

While the paper is well-written and presents a modular approach, the technical contribution is seen as incremental, largely extending existing methods in dynamic regret minimization and optimistic online learning. The experimental results are considered insufficiently convincing by some reviewers, with the performance improvement over existing methods being modest. Additionally, concerns were raised about algorithm-dependent bounds and the clarity of certain technical aspects.

**Additional Comments On Reviewer Discussion:**

The authors' responses have addressed some of the reviewer's concerns, but doubts still remain regarding the experiments and the technical contributions.

---

### Decision · Program_Chairs · 2025-01-22

Reject